# ANOMALY DETECTION EXPOSED: IMAGINING ANOMALIES WERE NORMAL

## ABSTRACT

Deep learning-based methods have achieved a breakthrough in image anomaly detection, but their complexity introduces a considerable challenge to understanding why an instance is predicted to be anomalous. We introduce a novel explanation method that generates multiple alternative modifications for each anomaly, capturing diverse concepts of anomalousness. Each modification is trained to be perceived as normal by the anomaly detector. The method provides a semantic explanation of the mechanism that triggered the anomaly detector, allowing users to explore "what-if scenarios." Qualitative and quantitative analyses across various image datasets demonstrate that applying this method to state-of-the-art anomaly detectors provides high-quality semantic explanations.

## 1 INTRODUCTION

Anomaly detection involves identifying patterns that deviate from normal behavior, the so-called *anomalies*. These anomalies can correspond to crucial actionable information in various domains such as medicine, manufacturing, surveillance, and environmental monitoring (Chandola et al., 2009; Hartung et al., 2023).

Recently, deep learning-based methods have shown tremendous success in anomaly detection (AD), reducing error rates to approximately 1% in numerous image benchmarks (Reiss et al., 2021; Deecke et al., 2021; Ruff et al., 2021; Liznerski et al., 2022). However, detectors based on deep learning lack the out-of-the-box interpretability of their traditional counterparts, making it difficult to understand the reasoning behind their predictions (Liznerski et al., 2021). Their lack of transparency is particularly concerning in sectors where safety is crucial and in situations where building trust is essential (Gupta et al., 2018; Montavon et al., 2018; Samek et al., 2020). Understanding modern anomaly detectors is a major challenge in contemporary AD and a necessary step before using AD in decision-making systems (Ruff et al., 2021).

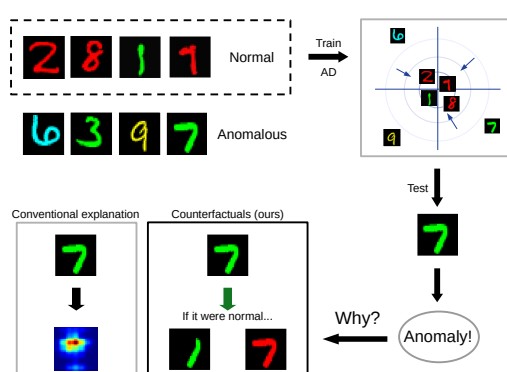

Figure 1: The figure illustrates the benefit of counterfactual explanation of anomaly detectors over traditional methods, using the Colored-MNIST dataset of handwritten digits in various colors. The normal data (top left) consist of red digits and instances of the digit one in any color. An example anomaly—a green seven—is shown on the right. Conventional explanation methods localize the anomaly within the image and highlight it on a heatmap (bottom left). In contrast, the proposed method transforms the anomaly into multiple counterfactuals.

Although feature-attribution techniques such as anomaly heatmaps (Liznerski et al., 2021; Gudovskiy et al., 2022; Roth et al., 2022) have been explored, they do not explain the underlying semantics of anomalies relevant to the decision-making of the detectors. In domains beyond AD, counterfactual explanation (CE) has emerged as a popular alternative. CE generates synthetic samples that change the model's prediction with minimal alterations to the original sample (Ghandeharioun et al., 2021; Abid et al., 2022). CEs are user-friendly and can provide explanations on a higher, semantic level.

In this paper, we propose the use of CE to explain anomaly detectors. To our knowledge, this paper presents the first study of CE in modern image AD based on deep learning. The AD setting comes with several considerable challenges. Anomalies can be rare and unlabeled in AD, making it difficult for deep generative models to synthesize realistic counterfactuals based on semantically meaningful concepts that are understandable to humans (Manduchi et al., 2024). Furthermore, normal samples can have limited diversity in AD, which complicates training deep generative models.

**Contributions** This paper introduces a novel unsupervised method for explaining image anomaly detectors using counterfactual examples. While previous approaches identify anomalous regions within images, the presented technique generates a set of counterfactual examples of each anomaly, capturing diverse disentangled aspects (see Figure 1). These counterfactual examples are created by transforming anomalous images into normal ones, guided by a specific aspect. The method provides semantic explanations of anomaly detectors, highlighting the higher-level aspects of an anomaly that triggered the detector. CE allows users to explore "what-if" scenarios (see Figure 1), improving the understanding of anomaly factors at an unprecedented level of abstraction. Qualitative and quantitative analyses across various image datasets show the effectiveness of the method when applied to state-of-the-art anomaly detectors. The code to reproduce the results and run the presented methods is included in the supplementary material.

## 2 RELATED WORK

In the past decade, research has increased on improving the interpretability and explainability of non-linear ML methods, particularly neural networks. This increase is driven by the growing use of ML in decision-making systems, where transparency of predictions is crucial and even legally mandated in many countries (Neuwirth, 2022). Here, we discuss key research articles relevant to our work. For a general overview of *explainable AI*, we refer to the survey by Linardatos et al. (2020).

**Explanation of image AD** Research in explainable image AD has primarily focused on feature attribution methods, pinpointing image areas that influence predictions. Some methods trace an importance score from the model output back to the pixels (Selvaraju et al., 2017; Zhang et al., 2018), others alter parts of the image and measure the impact on the model output. These alterations can include masking and noising (Fong & Vedaldi, 2017), blurring (Fong & Vedaldi, 2017), pixel values (Dhurandhar et al., 2018), or model outputs (Zintgraf et al., 2017). Some of these approaches have been applied to AD (Liznerski et al., 2021; Li et al., 2021; Wang et al., 2021). Several methods generate explanations using generative models or autoencoders, where the pixel-wise reconstruction error yields an anomaly heatmap (Baur et al., 2019; Bergmann et al., 2019; Dehaene et al., 2020; Liu et al., 2020; Venkataramanan et al., 2020). Others use fully convolutional architectures (Liznerski et al., 2021) or transfer learning (Defard et al., 2021; Roth et al., 2022). All of these methods identify regions within an image that influence the detector's prediction; however, they do not explain the detectors at a higher semantic level (Alqaraawi et al., 2020; Adebayo et al., 2018).

**Counterfactual explanation of neural networks on images** CE methods (Guidotti, 2022) identify the necessary changes in the input to alter the model prediction in a specific way. Unlike feature-attribution techniques, CE methods can explain predictions at a more sophisticated semantic level. Such explanations can provide profound insights that enhance comprehension of model behavior and align more closely with human cognitive processes (Pearl, 2009). Existing CE algorithms are designed primarily for supervised learning on tabular data (Wachter et al., 2017; Mothilal et al., 2020; Guidotti, 2022). A few studies have also explored the application of CE to image classification (Goyal et al., 2019; Ghandeharioun et al., 2021; Abid et al., 2022; Singla et al., 2023). DISSECT (Ghandeharioun et al., 2021) is particularly notable for its ability to generate multiple CEs with disentangled high-level concepts. However, to date, there is no existing work on the application of CE for image AD. Recent work explores CE for supervised image AD. Studies by Sanchez et al. (2022); Siddiqui et al. (2024); Ahamed et al. (2024) utilize diffusion models guided by text prompts or learnable conditions to generate normal counterparts of abnormal medical images. However, their approaches rely on supervised learning, fine-tuning pretrained diffusion models using both normal and ground-truth anomalies, framing the problem as a classification task. Wolleb et al. (2022) uses diffusion models with classifier guidance—trained in a supervised manner on normal and anomalous images—to transform diseased images into healthy ones. Fontanella et al. (2024) employ

a diffusion model trained exclusively on healthy brain images to generate saliency maps. However, they identify regions for counterfactual generation through supervised learning. Overall, none of the above approaches are designed for unsupervised anomaly detection, and they are constrained to particular types of images. Consequently, they are unsuitable for general image-AD.

**Counterfactual explanation of AD on shallow data** So far, CE methods for AD have been applied only to "shallow" data types, such as tables (Angiulli et al., 2023; Datta et al., 2022a; Han et al., 2023) or time series (Sulem et al., 2022; Cheng et al., 2022). These methods use knowledge graphs or structural causal models to generate counterfactuals for categorical features (Datta et al., 2022b; Han et al., 2023) or take advantage of temporal aspects (Sulem et al., 2022; Cheng et al., 2022). Some of these methods have been applied to fairness (Han et al., 2023) and algorithmic recourse (Datta et al., 2022a). None of the existing CE methods for AD are applicable to image data, nor are they capable of generating disentangled CEs. This capability is a unique characteristic of the proposed approach, which will be subsequently detailed.

## 3 METHODOLOGY

In this section, we formally present the proposed framework for generating counterfactuals in image AD using state-of-the-art generators. To the best of our knowledge, this approach is the first one to explain image AD using CE.

### 3.1 COUNTERFACTUAL EXPLANATIONS OF IMAGE AD

Our aim is to provide explanations for a given anomaly detector $\phi : \mathbb{R}^D \to [0, 1]$ that maps an image $x \in \mathbb{R}^D$ to an anomaly score $\alpha \in [0, 1]$. We define a CE for the detector $\phi$ and anomaly $\boldsymbol{x}^* \in \mathbb{R}^D$ (i.e., $\phi(\boldsymbol{x}^*) \gg 0$) as a modified sample $\bar{\boldsymbol{x}}^*$ with $\phi(\bar{\boldsymbol{x}}^*) \approx 0$ and $\|\bar{\boldsymbol{x}}^* - \boldsymbol{x}^*\|_1 \leq \epsilon$ for an $\epsilon \geq 0$. In other words, a CE must be normal according to $\phi$, while being minimally changed w.r.t. the original anomaly $\boldsymbol{x}^*$. Thus, CEs address the question: "What if the anomaly $\boldsymbol{x}$ were normal?", explaining the behavior of the anomaly detector at a high semantic level.

To produce such CEs for deep AD, we need to train a generator $G : \mathbb{R}^D \to \mathbb{R}^D$ to yield $G(\boldsymbol{x}^*) = \bar{\boldsymbol{x}}^*$. However, normal images can differ from anomalies in multiple ways, and thus multiple CEs may be required to adequately explain an anomaly. We want the generator to consider multiple categorical concepts $k \in \{1, \ldots, K\}$. Thus, the generator is now of the form $G : \mathbb{R}^D \times \{1, \ldots, K\} \to \mathbb{R}^D$ and is supposed to produce $G(\boldsymbol{x}^*, k) = \bar{\boldsymbol{x}}_k^*$ with $\|\bar{\boldsymbol{x}}_k^* - \bar{\boldsymbol{x}}_{k'}^*\|_1 \geq \epsilon'$.

The same data $\{(\boldsymbol{x}_0, y_0), \ldots, (\boldsymbol{x}_n, y_n)\}$ can be used for training both $\phi$ and $G$. Here, $y_i = 0$ denotes normal samples, while $y_i = 1$ represents anomalies. Note that in the AD setting, the training labels $y_i$ are typically unknown and the majority of samples are assumed to be normal.

### 3.2 DISENTANGLED COUNTERFACTUAL EXPLANATIONS

Outside the domain of AD, Ghandeharioun et al. (2021) have proposed Disentangled Simultaneous Explanations via Concept Traversal (DISSECT) to create CEs. DISSECT produces sequences of CEs with increasing impact on a classifier's output. The proposed approach for CE of image anomaly detectors is based on this idea.

We modify the generator $G : \mathbb{R}^D \times [0, 1] \times \{1, \ldots, K\} \to \mathbb{R}^D$ to also consider a target anomaly score $\alpha$, aiming for the trained $G$ to produce a sample with an anomaly score of approximately $\alpha$. Following DISSECT, we train $G$ as a concept-disentangled GAN Goodfellow et al. (2020). To this end, we define a discriminator $D : \mathbb{R}^D \to [0, 1]$ and a concept classifier $R : \mathbb{R}^D \times \mathbb{R}^D \to [0, 1]^K$. $D$ is trained to distinguish between generated $\bar{\boldsymbol{x}}_{\alpha,k} = G(\boldsymbol{x}, \alpha, k)$ and true samples from the dataset, encouraging *realistic* outcomes. $R$ classifies the concept $k$ for a sample $\bar{\boldsymbol{x}}_{\alpha,k}$, encouraging the generated samples to be *concept-disentangled* on a semantic level. Further losses encourage the generator to incur *minimal changes* on the original sample $\boldsymbol{x}$ and to yield target anomaly scores $\alpha$ (i.e., $\phi(\bar{\boldsymbol{x}}_{\alpha,k}) \approx \alpha$).

The proposed method's objective summarizes to

$$\min_{G,R} \max_{D} \lambda_{gan} \left( L_D(D) + L_G(G) \right) + \lambda_\phi L_\phi(G) + \lambda_{rec} L_{rec}(G) + \lambda_{rec} L_{cyc}(G) + \lambda_r L_{con}(G, R),$$

where $L_\phi(G)$ encourages for $\bar{\boldsymbol{x}}_{\alpha,k}$ an anomaly score of $\alpha$:

$$L_\phi(G) = \alpha \log\big(\phi(\bar{\boldsymbol{x}}_{\alpha,k})\big) + \big(1 - \alpha\big) \log\big(1 - \phi(\bar{\boldsymbol{x}}_{\alpha,k})\big).$$

The losses $L_D(D)$ and $L_G(G)$ can be any discriminative and generative GAN losses, respectively. We specifically experimented with the spectrally normalized loss $L_G(G) = -D(\bar{\boldsymbol{x}}_{\alpha,k})$ Miyato et al. (2018) and the hinge loss Miyato & Koyama (2018):

$$L_D(D) = -\min(0, -1 + D(\boldsymbol{x})) - \min(0, -1 - D(\bar{\boldsymbol{x}}_{\alpha,k})).$$

The loss $L_{rec}(G) = \|\boldsymbol{x} - G(\boldsymbol{x}, \phi(\boldsymbol{x}), k)\|_1$ makes $G$ reconstruct $\boldsymbol{x}$ for every concept $k$, when conditioned on $\boldsymbol{x}$ and its "true" anomaly score $\phi(\boldsymbol{x})$. This ensures that $G$ remains unchanged when the sample already has the targeted anomaly score, overall encouraging minimal changes.

Similarly, the "cycle consistency loss" Zhu et al. (2017), $L_{cyc}(G) = \|\boldsymbol{x} - \tilde{\boldsymbol{x}}_{\alpha,k}\|_1$, where $\tilde{\boldsymbol{x}}_{\alpha,k} = G(\bar{\boldsymbol{x}}_{\alpha,k}, \phi(\boldsymbol{x}), k)$, encourages $G$ to recreate the sample $\boldsymbol{x}$, when targeting its true anomaly score $\phi(\boldsymbol{x})$ and being conditioned on any generated sample $\bar{\boldsymbol{x}}_{k,\alpha}$ based on $\boldsymbol{x}$. It encourages minimal changes because the generator needs to be able to revert any change of $\boldsymbol{x}$.

$L_{con}(G, R)$ drives $G$ to produce disentangled concepts:

$$L_{con}(G, R) = \mathbb{C}\Big(k, R\big(\boldsymbol{x}, \bar{\boldsymbol{x}}_{\alpha,k}\big)\Big) + \mathbb{C}\Big(k, R\big(\bar{\boldsymbol{x}}_{k,\alpha}, \tilde{\boldsymbol{x}}_{\alpha,k}\big)\Big),$$

where $\mathbb{C}$ denotes the cross entropy loss.

In summary, the losses encourage the generated samples $\bar{\boldsymbol{x}}_{\alpha,k}$ to be semantically distinguishable for different concepts $k$ while having an anomaly score of $\alpha$ according to $\phi$ and undergoing minimal changes with respect to the original $\boldsymbol{x}$. This results in a disentangled set of $K$ counterfactual examples for an anomaly $\boldsymbol{x}^*$ with $\{G(\boldsymbol{x}^*, 0, 1), \ldots, G(\boldsymbol{x}^*, 0, K)\}$. Furthermore, the generator can also produce pseudo anomalies $G(\boldsymbol{x}, \alpha, K)$ when $\phi(\boldsymbol{x}) \approx 0$ and $\alpha \gg 0$, which can help $G$ in learning how to turn anomalies into normal samples, when included in $L_\phi$.

**CE using diffusion models**  We also adapt DiffEdit (Couairon et al., 2023) to generate counterfactual explanations. DiffEdit modifies the LAION-5B pre-trained text-conditional latent diffusion model known as Stable Diffusion (Rombach et al., 2022) to semantically edit images. Let $A_\mathcal{E} : \mathbb{R}^D \to \mathbb{R}^\Delta$ and $A_\mathcal{D} : \mathbb{R}^\Delta \to \mathbb{R}^D$ denote the encoder and decoder of the autoencoder used in Stable Diffusion. From a high-level perspective, the DiffEdit model can be defined as $\psi : \mathbb{R}^{\Delta \times T} \to \mathbb{R}^\Delta$ where $T$ denotes the output dimension of the word embedding model. For an image $\boldsymbol{x} \in \mathbb{R}^D$, we retrieve a semantically modified version $\hat{\boldsymbol{x}}$ controlled by the text prompt $t$ via $\hat{\boldsymbol{x}} = A_\mathcal{D}(\psi(A_\mathcal{E}(\boldsymbol{x}), t))$. For more details, refer to the paper (Couairon et al., 2023). We incorporate DiffEdit into the proposed framework by training the generator on its latent output. That is, we redefine the generator $G(\boldsymbol{x}, \alpha, k) = A_\mathcal{D}\left(G'(\psi(A_\mathcal{E}(\boldsymbol{x}), t), \alpha, k)\right)$ with $G' : \mathbb{R}^\Delta \times [0, 1] \times \{1, \ldots, K\} \to \mathbb{R}^\Delta$. The text prompt $t$ is set to the normal class label (e.g., "cat" for cats being normal). We train the generator $G$ (i.e., the parameters of $G'$) as described before. Incorporating DiffEdit as described here allows one to apply the proposed framework to higher-resolution images, where training from scratch quickly becomes infeasible.

### 3.3 DEEP ANOMALY DETECTION

The proposed CE framework is general and can be applied to any anomaly detector that produces real-valued anomaly scores. In this paper, we specifically study three state-of-the-art anomaly detectors that are reviewed below.

**DSVDD**  One of the first deep approaches to AD is Deep Support Vector Data Description (DSVDD) Ruff et al. (2018). Similar to many AD methods, DSVDD is unsupervised, employing an unlabeled corpus of data for training. DSVDD trains a neural network $\phi_\theta : \mathbb{R}^D \to \mathbb{R}^d$ with parameters $\theta$ to map the training data $\boldsymbol{x}_1, \ldots, \boldsymbol{x}_n \in \mathbb{R}^D$ into a semantic space $\mathbb{R}^d$, where it can be enclosed by a minimal volume hypersphere: $\min_\theta \sum_{i=1}^n \|\phi_\theta(\boldsymbol{x}_i) - \boldsymbol{c}\|^2$. In contrast to shallow SVDD Tax & Duin (2004), the hypersphere center $\boldsymbol{c} \in \mathbb{R}^d$ is first randomly initialized and then kept fixed while training. DSVDD trains the network to make normal data cluster tightly in the semantic space. Anomalies

will have a larger distance from the center. The distance is used as the anomaly score. Since the CE generator requires bounded anomaly scores, we slightly adjust the DSVDD objective to:

$$\min_{\theta} \sum_{i=1}^{n} \frac{||\phi_{\theta}(\boldsymbol{x}_i) - \boldsymbol{c}||^2}{1 + ||\phi_{\theta}(\boldsymbol{x}_i) - \boldsymbol{c}||^2}.$$

**Outlier Exposure**  AD has traditionally been approached as an unsupervised learning problem due to insufficient training data to represent the diverse anomaly class, which encompasses *everything different* from the normal data. However, Hendrycks et al. (2019a) showed that *Outlier Exposure* (OE)—using a large unstructured collection of natural images as example anomalies during training— consistently outperforms purely unsupervised AD methods across various image-AD benchmarks. These auxiliary data are called OE samples. It has been found that training a Binary Cross Entropy (BCE) loss to differentiate normal data from OE samples is competitive for most image-AD tasks. We use the OE samples both for training the detector's network $\phi$ and the generator $G$. The generator $G$ is thus trained on a more diverse training set, including additional presumably anomalous OE samples.

**Hypersphere Classification**  Although OE performs well in many benchmarks, there are still scenarios where OE samples do not adequately represent anomalies, especially when the normal data are not natural images Liznerski et al. (2022). To address this problem, the community has developed *semi-supervised* AD methods Görnitz et al. (2014); Ruff et al. (2020). One of the most competitive semi-supervised AD techniques is *HyperSphere Classification* (HSC) Liznerski et al. (2022). The authors find that combining it with OE makes the AD more robust to the selection of OE data. The HSC loss is a semi-supervised modification of the DSVDD loss:

$$\frac{1}{n} \sum_{i=1}^{n} y_i \cdot h\left(\phi_{\theta}(\boldsymbol{x}_i)\right) - (1 - y_i) \log\left(1 - \exp\left(-h\left(\phi_{\theta}(\boldsymbol{x}_i)\right)\right)\right),$$

where $h$ is the Pseudo-Huber loss $h(\boldsymbol{z}) = \sqrt{||\boldsymbol{z}||^2 + 1} - 1$. We employ HSC's original objective but modify the anomaly score from $h\left(\phi_{\theta}(\boldsymbol{x}_i)\right)$ to $1 - \exp(-h\left(\phi_{\theta}(\boldsymbol{x}_i)\right))$, again obtaining bounded anomaly scores for training the proposed counterfactual generator.

## 4 EXPERIMENTS

In this section, we empirically assess the capabilities of CEs for deep AD. The evaluation provides qualitative (Section 4.2) and quantitative (Section 4.3) evidence of the superiority of the proposed CEs over their traditional counterparts. Notably, the experiments expose a previously unreported bias of supervised classifiers when used in the AD setting (Section 4.4).

### 4.1 EXPERIMENTAL DETAILS

We describe the considered datasets, the experimental setup, and the implementation of the method.

**Datasets**  We evaluate the proposed approach on the following datasets:
- MNIST (Deng, 2012) is a dataset of grayscale handwritten digits with a class for each digit. Following Liznerski et al. (2021), we use EMNIST (Cohen et al., 2017) as OE.
- Colored-MNIST, where for each sample in MNIST, copies are created in seven colors (red, yellow, green, cyan, blue, pink, and gray). We employ a colored version of EMNIST as OE.
- CIFAR-10 (Krizhevsky et al., 2009) is a dataset of natural images with ten classes. Previous works used 80 Mio. Tiny Images as OE (Hendrycks et al., 2019b). Since this dataset has been withdrawn due to offensive data Birhane & Prabhu (2021), we instead use the disjunct CIFAR-100 dataset as OE, which yields approximately the same performance (here 96.0% average AuROC, as reported in Table 8, vs. 96.1% AuROC in Liznerski et al. (2022)).
- GTSDB Houben et al. (2013) is a dataset of German traffic signs. We use CIFAR-100 as OE.
- We introduce ImageNet-Neighbors (INN), a subset of ImageNet-1k (Russakovsky et al., 2015) designed for anomaly detection (AD) tasks. INN comprises multiple AD setups; in each setup, one ImageNet-1k class is considered normal, and the ten most semantically similar classes, based on

the Wu-Palmer similarity metric (Wu & Palmer, 1994), are defined as ground-truth test anomalies. For outlier exposure (OE), we use the disjoint ImageNet-21k dataset.

**Experimental Setup** Following previous work on image-AD Ruff et al. (2018); Golan & El-Yaniv (2018); Hendrycks et al. (2019a;b); Ruff et al. (2020); Tack et al. (2020); Ruff et al. (2021); Liznerski et al. (2021; 2022), we convert several multi-class classification datasets into AD benchmarks. This is achieved by defining a subset of the classes to be normal and using the remaining classes as ground-truth anomalies during testing. When only one class is considered normal, this approach is known as one vs. rest. In addition to investigating one vs. rest, we also explore a variation in which multiple classes are normal. This setting emulates a multifaceted normal class that includes different notions of normality. Since our method disentangles multiple aspects of the normal data, we hypothesize that it possesses the capability to capture these diverse facets of normality. Finally, we consider the special INN setup, as described above, where we have particular ground-truth anomalies per normal class. Our experiments focus on semantic image-AD rather than low-level AD, where anomalies are defects instead of out-of-class (such as in datasets like MVTec-AD (Bergmann et al., 2019)). We include further reasoning for this and an ablation study for CEs on MVTec-AD in Appendix C.

For both the MNIST and CIFAR-10 datasets, we construct 30 distinct scenarios: ten scenarios wherein each individual class serves as the normal data, and an additional 20 scenarios featuring various combinations of classes as normal. For the Colored-MNIST dataset, we define seven normal-class scenarios through combinations of colors and digits. We consider ten different normal-class sets for the GTSDB dataset. For ImageNet-Neighbors, we consider five different normal classes. For each scenario and several random seeds, we train an AD model and a CE generator. For INN, we train a generator based on DiffEdit, as described in the methodology section, while the other scenarios train a GAN from scratch. Details of all scenarios are provided in Appendix G. Our quantitative analysis reports results averaged over all scenarios and multiple seeds. Detailed quantitative results for each scenario are in Appendix G and a collection of further qualitative results in Appendix H.

**Implementation Details** In our experiments, we generate and compare CEs using three state-of-the-art deep AD methods: BCE, HSC, and DSVDD (see Section 3.3). We employ conventional convolutional neural networks with up to five layers for the AD methods. The concept classifier is a small ResNet He et al. (2016) with two blocks. Both the discriminator and generator are wide ResNets Zagoruyko & Komodakis (2016) with four blocks. The $\lambda$ parameters in our loss (Section 3.2) are set to reasonable values that have been found to perform well across all settings. The hyperparameters of the AD methods are chosen as in previous work Ruff et al. (2018); Liznerski et al. (2022). The epochs and augmentation are slightly reduced for faster training. A description of all hyperparameters and network architectures is given in Appendix E for both the CE generator and AD methods.

## 4.2 QUALITATIVE RESULTS

In this section, we present qualitative examples of CEs on four datasets, demonstrating the benefit of using CE for AD over traditional explanation methods.

### 4.2.1 COUNTERFACTUALS CAN EXPLAIN WHY IMAGES ARE PREDICTED ANOMALOUS

**Colored-MNIST** Figure 2 shows the counterfactual explanations for Colored-MNIST, when the normal class is formed from the instances of the digit one and digits colored cyan. We observe that the CEs generated to explain the BCE detector align well with our expectation. The proposed method transforms the anomalies into ones without changing the color, or their color is changed to cyan without changing the digit. Both modifications are minimal alterations of the anomaly, transforming its appearance to normality in two distinct ways. The CEs of the HSC method also mostly correspond to normal samples, as expected. However, in some cases,

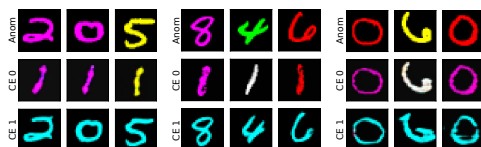

(a) BCE (OE)    (b) HSC (OE)    (c) DSVDD

Figure 2: CEs for the Colored-MNIST dataset, with cyan digits and the digit one serving as the normal class. The first row shows anomalous images, and the next two rows present their corresponding CEs using two different concepts. The CEs of BCE and HSC appear normal and realistic for each concept.

both the color and the digit is changed, resulting in unnecessary changes. We found that this behavior represents a local optimum of the objective of our method, highlighting the inherent difficulty of the unsupervised generation of CEs for AD. The CEs created to explain the DSVDD detector perform the least effectively. They tend to appear normal for one concept but often fail for the other concept. This behavior may be attributed to DSVDD's limited ability to detect anomalies, when compared with the more competitive BCE and HSC detectors, which have the advantage of having access to OE.

**MNIST** In Figure 3, a single digit (seven) or multiple digits (eight and nine) are considered normal.

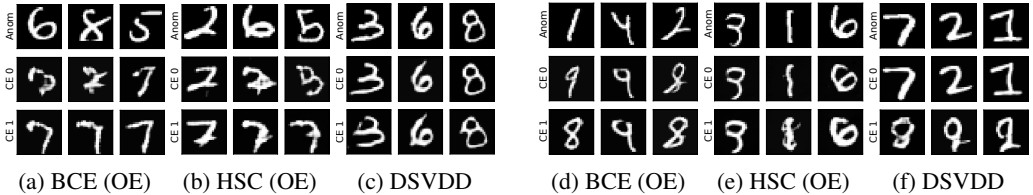

(a) BCE (OE)  (b) HSC (OE)  (c) DSVDD  (d) BCE (OE)  (e) HSC (OE)  (f) DSVDD

Figure 3: Examples of CEs for MNIST, (a-c) with the digit seven as the normal class, and (d-f) with digits eight and nine forming the normal class. The first row shows anomalous images, the other two rows show CEs using two different concepts. CEs of BCE and HSC in (a,b) are variations of seven and thus represent intuitive counterfactuals. CEs of BCE and DSVDD in (d) resemble normal eights or nines for the second concept.

When the single digit seven is considered normal, the CEs of BCE and HSC are meaningful: the anomalies are transformed into variations of seven. Notably, when the digits eight and nine are considered normal, some anomalies are turned into eights, and others into nines. This observation confirms our hypothesis that our method can correctly reveal diverse notations of normality in multifaceted normal data. As expected, the CEs of DSVDD are generally worse.

**GTSDB** Figure 4 shows the proposed CEs for the GTSDB dataset, when speed signs are taken as a normal class. We refer to Appendix H for more experimental results using other normal scenarios with similar findings. The CEs of BCE and HSC show well-disentangled normal traffic signs, obtained from anomalous ones. For instance, the CE of BCE changes the "80km/h restriction ends" sign into a "80km/h limit" sign, which is a minimal intervention to make the sample appear normal. Note that all triangular

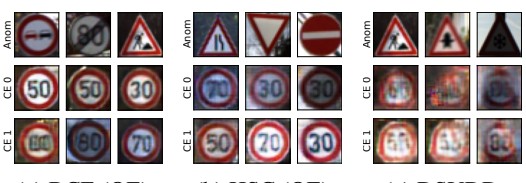

(a) BCE (OE)  (b) HSC (OE)  (c) DSVDD

Figure 4: CEs for GTSDB with speed signs forming the normal class. The first row shows anomalous images, the other two rows disentangled CEs.

anomalies are changed to circles. The CEs show that the shape is an important feature for the detector to rate anomalousness.

**CIFAR-10** Especially for BCE, the CEs for CIFAR-10 in Figure 5 represent intuitive normal samples (ships) that retain the anomalous object's color to incur minimal changes on the anomaly. As there is only one single normal class, the CEs generated for HSC and BCE primarily disentangle the concepts by changing the background. Typically, ships are depicted floating on water, which may vary in color. CEs for DSVDD are generally worse, revealing weaknesses of DSVDD as discussed in Appendix B. We refer to Appendix H for more experimental results using other normal classes, demonstrating that CEs exhibit a similar behavior for combinations of classes forming normality.

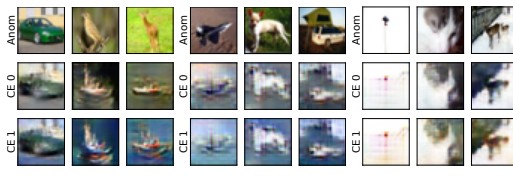

(a) BCE (OE)  (b) HSC (OE)  (c) DSVDD

Figure 5: Examples of CEs for CIFAR-10, when images of ships are normal. The first row shows anomalous images, the other two rows present CEs using two different concepts. The CEs of BCE and HSC display normal ships, varying the background for successful disentanglement while keeping the object's color to avoid unnecessary changes.

**ImageNet-Neighbors** Figure 6 shows CEs for the INN dataset when zebras are normal. The ground-truth anomalies are "similar" animals, ranging from horses and boars to armadillos. Since DSVDD does not perform competitively, we show results for BCE and HSC only. The CEs depict zebras while keeping the general pose and background of the anomalous animal. For disentanglement, the CEs vary the color scheme, which apparently the detectors perceive as normal. The CEs for the second concept for HSC are dark and, while still showing zebras, perturb the image with green and orange patterns. Interestingly, the HSC detector assigns lower anomaly scores to the CEs for the second concept.

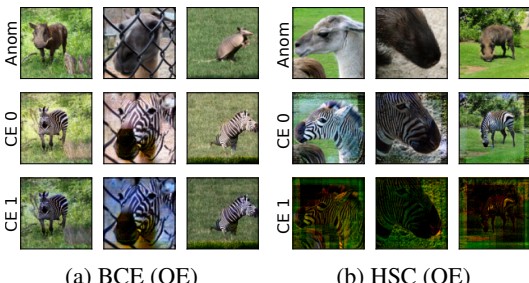

(a) BCE (OE)      (b) HSC (OE)

Figure 6: Examples of CEs for INN, where images of zebras are considered normal. The first row shows anomalous images, the other two rows present CEs using two different concepts.

#### 4.2.2 COUNTERFACTUALS CAN EXPLAIN WHY IMAGES ARE PREDICTED ANOMALOUS—*even when feature attribution fails*

Here, we demonstrate the advantage of the proposed CEs over conventional explanations that attribute features to localize anomalies. Figure shows 7 (a) CEs generated with our method and (b) heatmaps for the corresponding anomalies generated with FCDD Liznerski et al. (2021).

FCDD's heatmaps explain only spatial aspects of the anomalies: FCDD highlights the horizontal bar in digit seven, the circle in digit nine, and all of digit eight. These spatial aspects of anomalies are also explained by the CEs created for the first concept, where the anomalies are turned into the digit one. However, FCDD's heatmaps fail to identify the color as being anomalous, whereas the proposed CEs capture this aspect with their second concept, where the anomalies are colored red, making them look normal. This demonstrates that CEs can provide more holistic explanations of anomalies.

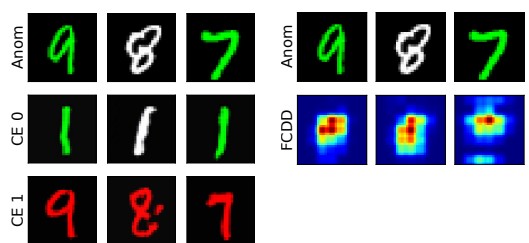

(a) Counterfactuals      (b) Heatmaps with FCDD

Figure 7: The first row shows anomalies from Colored-MNIST, with red digits and the digit one forming the normal class. The other rows show (a) corresponding CEs for two concepts, and (b) anomaly heatmaps generated with FCDD Liznerski et al. (2021). The CEs explain the anomaly detector that perceives anomalies turned red or into one as normal, while heatmaps just highlight the difference to one.

### 4.3 QUANTITATIVE RESULTS

This section presents a quantitative analysis of the CEs, assessing their normality, realism, disentanglement, and suitability for training anomaly detectors in terms of various metrics based on AuROC, FID, and accuracy. These metrics are described in detail in Appendix D.

#### 4.3.1 THE COUNTERFACTUALS APPEAR AS NORMAL

An important attribute for any CE in deep AD is that it must be perceived as normal by the anomaly detector. To evaluate this quality criterion, we compare the anomaly scores of the normal test samples with those of the generated CEs in terms of AuROC. Ideally, the AuROC should approach 50%, indicating that CE and normal samples are indistinguishable. As shown in Table 1, the AuROC is indeed very close to

Table 1: The AuROC of normal test data vs. CEs. The CEs appear entirely normal for values $\leq 50\%$.

|  | Datasets | Methods | | |
|---|---|---|---|---|
|  |  | BCE OE | HSC OE | DSVDD |
| Single normal class | MNIST | $72.0 \pm 4.0$ | $80.8 \pm 5.3$ | $75.2 \pm 9.2$ |
|  | CIFAR-10 | $47.5 \pm 10.0$ | $49.9 \pm 4.4$ | $54.6 \pm 3.4$ |
|  | INN | $69.1 \pm 18.1$ | $67.9 \pm 13.2$ | $\times$ |
| Multiple normal classes | C-MNIST | $55.6 \pm 1.5$ | $55.8 \pm 4.7$ | $61.5 \pm 4.3$ |
|  | MNIST | $78.1 \pm 4.1$ | $82.1 \pm 3.8$ | $73.4 \pm 6.5$ |
|  | CIFAR-10 | $49.0 \pm 8.5$ | $44.4 \pm 6.7$ | $50.7 \pm 3.3$ |
|  | GTDSB | $50.2 \pm 8.0$ | $48.6 \pm 14.4$ | $53.1 \pm 4.8$ |

50% on CIFAR-10, GTSDB, and Colored-MNIST (here abbreviated as C-MNIST), underlining that the detector perceives the CEs as normal. Only on MNIST and INN, some of the CEs appear anomalous. This might be due to the enforced disentanglement that produces diverse samples despite a limited variety of possible normal variations.

### 4.3.2 THE COUNTERFACTUALS CAN BE USED TO TRAIN AN ANOMALY DETECTOR EFFECTIVELY

If the CEs resemble normal images, they can serve as viable normal training samples. We retrain the AD methods using CEs instead of the normal training set and report the AuROC for normal vs. anomalous test samples in Table 2a. The results show that the CEs are effective normal training samples, as the AuROC values are mostly well above the chance level of 50%.

Table 2: AuROC of normal vs. anomalous test samples when (a) the AD is trained with the normal training set being substituted with CEs and (b) the AD is trained with the usual normal training set.

(a) AD AuROC with the CEs as normal training data.

|  | Datasets | Methods | | |
| --- | --- | --- | --- | --- |
|  |  | BCE OE | HSC OE | DSVDD |
| Single normal class | MNIST | $91.3 \pm 4.6$ | $85.6 \pm 9.2$ | $46.2 \pm 10.5$ |
|  | CIFAR-10 | $59.0 \pm 6.1$ | $54.8 \pm 2.6$ | $50.8 \pm 3.2$ |
|  | INN | $59.2 \pm 5.8$ | $53.0 \pm 11.0$ | $\times$ |
| Multiple normal classes | C-MNIST | $80.6 \pm 4.5$ | $81.7 \pm 4.8$ | $59.9 \pm 8.4$ |
|  | MNIST | $62.2 \pm 13.2$ | $54.7 \pm 9.9$ | $41.6 \pm 4.5$ |
|  | CIFAR-10 | $58.7 \pm 4.6$ | $53.1 \pm 1.8$ | $49.7 \pm 4.1$ |
|  | GTDSB | $90.1 \pm 5.3$ | $89.9 \pm 5.1$ | $58.4 \pm 7.0$ |

(b) AD AuROC with the proper normal training set.

|  | Datasets | Methods | | |
| --- | --- | --- | --- | --- |
|  |  | BCE OE | HSC OE | DSVDD |
| Single normal class | MNIST | $97.7 \pm 1.5$ | $97.6 \pm 1.6$ | $78.8 \pm 8.6$ |
|  | CIFAR-10 | $96.0 \pm 2.5$ | $95.9 \pm 2.5$ | $55.4 \pm 4.7$ |
|  | INN | $93.6 \pm 5.7$ | $92.6 \pm 6.7$ | $\times$ |
| Multiple normal classes | C-MNIST | $97.1 \pm 1.0$ | $95.7 \pm 2.3$ | $76.9 \pm 6.5$ |
|  | MNIST | $93.5 \pm 2.8$ | $92.9 \pm 3.3$ | $75.4 \pm 7.1$ |
|  | CIFAR-10 | $93.8 \pm 2.7$ | $94.0 \pm 2.7$ | $52.6 \pm 3.6$ |
|  | GTDSB | $94.3 \pm 4.7$ | $93.0 \pm 5.6$ | $58.2 \pm 6.7$ |

The AD methods significantly outperform a random detector when trained with CEs, affirming their viability as normal samples. A notable exception is DSVDD, a method that does not utilize OE and struggles when trained purely with CEs. Table 2b shows the AuROC values of the models when trained with the proper normal training set.

### 4.3.3 THE COUNTERFACTUALS ARE REALISTIC

To assess the realism of the CEs, we compute the FID between CEs and normal test samples. For an intuitive score, we normalize the FID for CEs by dividing by the FID between normal and anomalous test samples. The normalized FID is 100% if the CEs are equally realistic as the anomalies. Details are provided in Appendix D. We found that a normalized FID of 50 to 100% is a reasonable target for expressive CEs. If the CEs became too similar to the normal data distribution, they would not be valid counterfactuals, as they would not retain non-anomalous features from the anomalies. Table 3 displays the normalized FID scores. The CEs for BCE and HSC are mostly as realistic as the anomalies. On MNIST, INN and Colored-MNIST, the CEs are even more realistic than the anomalies. As CEs for DSVDD tend to reconstruct anomalies, their realism is also reasonable.

Table 3: Normalized FID scores for the CEs. Most of the CEs are as realistic as the anomalies, which are also realistic since they follow the general data distribution (e.g., are digits in case of MNIST).

|  | Datasets | Methods | | |
| --- | --- | --- | --- | --- |
|  |  | BCE OE | HSC OE | DSVDD |
| Single normal class | MNIST | $43 \pm 8.1$ | $68 \pm 14.6$ | $100 \pm 8.8$ |
|  | CIFAR-10 | $116 \pm 20.8$ | $300 \pm 90.0$ | $116 \pm 12.0$ |
|  | INN | $85.0 \pm 28.6$ | $85.4 \pm 24.6$ | $\times$ |
| Multiple normal classes | C-MNIST | $56 \pm 12.4$ | $95 \pm 30.5$ | $83 \pm 8.7$ |
|  | MNIST | $78 \pm 26.0$ | $96 \pm 25.0$ | $100 \pm 10.7$ |
|  | CIFAR-10 | $103 \pm 27.9$ | $254 \pm 69.7$ | $110 \pm 10.0$ |
|  | GTDSB | $110 \pm 101.8$ | $95 \pm 73.5$ | $131 \pm 118.1$ |

### 4.3.4 THE COUNTERFACTUALS CAPTURE MULTIPLE DISENTANGLED ASPECTS

Here we show that, for each anomaly, our method generates concept-disentangled CEs. Recall that the concept classifier is trained to predict the concept of each CE (see Section 3). Consequently, we have a metric for assessing the disentanglement of the generated samples. We present the accuracy of this concept classifier on test data in Table 4.

Our models demonstrate a consistent ability to disentangle concepts effectively, with the exception of DSVDD, which has suboptimal AD performance, making it difficult to provide explanations in general. In particular, disentanglement is effective even in the case where just one class is considered normal. On CIFAR-10 the generator exploits the background, on INN the color scheme, and on MNIST it generates disentangled variants of digits. We hypothesize that this strong disentanglement is the reason behind the CEs appearing less normal for MNIST.

Table 4: The accuracy of the concept classifier for the generated CEs.

|  | Datasets | Methods | | |
|---|---|---|---|---|
|  |  | BCE OE | HSC OE | DSVDD |
| Single normal class | MNIST | $94.3 \pm 3.9$ | $90.8 \pm 4.8$ | $77.5 \pm 14.1$ |
|  | CIFAR-10 | $93.0 \pm 4.3$ | $98.8 \pm 3.2$ | $97.1 \pm 2.9$ |
|  | INN | $97.0 \pm 5.4$ | $98.9 \pm 1.1$ | $\times$ |
| Multiple normal classes | C-MNIST | $99.4 \pm 1.3$ | $98.9 \pm 2.0$ | $98.0 \pm 3.0$ |
|  | MNIST | $93.8 \pm 5.1$ | $85.7 \pm 9.6$ | $81.6 \pm 11.3$ |
|  | CIFAR-10 | $86.2 \pm 7.5$ | $98.9 \pm 2.4$ | $92.2 \pm 4.2$ |
|  | GTDSB | $98.8 \pm 0.8$ | $94.0 \pm 8.4$ | $93.4 \pm 4.5$ |

### 4.4 COUNTERFACTUALS REVEAL A PREVIOUSLY UNREPORTED CLASSIFIER BIAS IN DEEP AD

In this section, we present a scientific finding: classifiers may be biased when trained for deep AD. The hypothesis of "classification bias," suggesting supervised classifiers underperform when trained with limited and biased anomaly subsets Ruff et al. (2020), remains insufficiently investigated. To test this hypothesis, we train a supervised classifier on Colored-MNIST, aiming to distinguish between a normal set (red digits and the digit one) and a subset of the ground-truth anomalies, specifically all blue anomalies. We select a subset of the anomalies for training to simulate a realistic scenario in which one has no access to all variations of the ground-truth anomalies. A key requirement in AD is the model's ability to identify all forms of unseen anomalies. The classifier bias becomes apparent as the AuROC of normal test samples vs. ground-truth anomalies decreases from 98 for

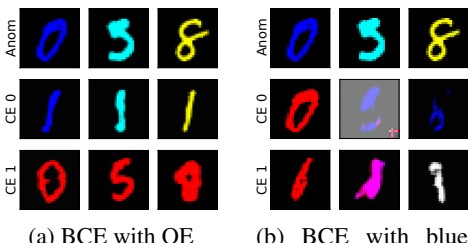

(a) BCE with OE    (b) BCE with blue anomalies

Figure 8: The first row shows anomalies for Colored-MNIST with red digits and the digit one forming the normal class. The other two rows present CEs of BCE trained with OE in (a) and of a classifier trained with only blue anomalies in (b). The generator's inability to generate normal-looking CEs for anomalies other than blue suggests that the classifier in (b) is biased.

BCE with OE (unsupervised) to 75 for supervised BCE. Our CEs further illuminate this phenomenon (see Figure 8). While our explanation for the AD method with OE in (a) indicates that anomalies should be transformed into red or digit one to appear normal, they depict a different picture for the supervised classifier in (b). Here, only for the blue anomalous zero, which is seen during training, the CEs roughly show intuitive normal versions of the anomaly. For other unseen anomalies, such as the cyan five or yellow eight, the explanations do not show intuitive normal images. This suggests that the classifier is biased towards detecting blue anomalies and fails to generalize to other colors not present in the training set. This underlines the need for specialized AD methods (e.g., using OE or semi-supervised objectives) because they are less prone to bias.

## 5 CONCLUSION

This paper introduced a novel method that can interpret image anomaly detectors at a semantic level. This is achieved by modifying anomalies until they are perceived as normal by the detector, creating instances known as counterfactuals. We found that counterfactuals can provide a deeper, more nuanced understanding of image anomaly detectors, far beyond the traditional feature-attribution level. Extensive experiments across various image benchmarks and deep anomaly detectors demonstrated the efficacy of the proposed approach. This research marks a paradigm shift and a significant departure from the more superficial interpretation of anomaly detectors using feature attribution, enhancing our understanding of detectors on a more abstract, semantic level. This may be a substantial milestone in the pursuit of more transparent and accountable AD systems.

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

## A  BROADER IMPACT

As an explanation technique, our method naturally aids in making deep AD more transparent. It may reveal biases in the model (see Section 4.4) and improve trustworthiness. For example, it may reveal a social bias when a portrait of a person is labeled anomalous due to race or gender. In this scenario, our method might generate CEs where merely the skin color has been changed. Applying our method can prevent a harmful deployment of such an AD model.

## B  LIMITATIONS OF OUR APPROACH

In the main paper, we proposed a method to generate counterfactual explanations (CEs) for deep anomaly detection (AD). As seen in Section 4, the quality of the generated counterfactual explanations relies on the performance of the AD model. DSVDD without OE Ruff et al. (2018) performs weakly on some image datasets. Consequently, CEs for DSVDD are often not very intuitive and sometimes collapse to a mere reconstruction of the anomaly. This happens because DSVDD struggles to recognize an anomaly and thus assigns a low anomaly score to it. Our method doesn't have a reason to change an anomaly to turn it normal for DSVDD. Another limitation of our method is that the generator might change more than necessary to turn the anomaly normal, thereby falling into a local optimum of the overall objective. Learning to balance the objectives of our method in an unsupervised manner is challenging, especially given the limited variety and amount of normal samples. Future work may improve upon this.

## C  COUNTERFACTUAL EXPLANATIONS OF DEFECTS

In the main paper, we did not include experiments on datasets such as MVTec-AD, where anomalies are subtle modifications of normal samples (e.g., cracked hazelnuts for healthy hazelnuts being normal) rather than being out of class. Such datasets are not interesting in the context of high-level explanations. Contrary to usual assumptions in AD, where anomalies are *everything*, which is not normal, in MVTec-AD there is a very precise definition of anomalousness and only one specific way to turn anomalies normal (i.e., by removing the defect). CEs would not help in understanding the model. Hence, we focus on the well-established and important semantic image-AD setting.

To visualize why CEs are not a useful tool for explaining low-level AD, we trained our proposed method from scratch with a single concept on several classes of MVTec-AD. Figure 9 shows some generated CEs for the classes bottle, grid, hazelnut, metal nut, screw, tile, and wood. Mostly, the CEs are high-quality: realistic and normal. However, they do not help us to understand the behavior of the model. They simply show the sample with the defect removed, which is a trivial explanation of the anomaly but does not explain the anomaly detector.

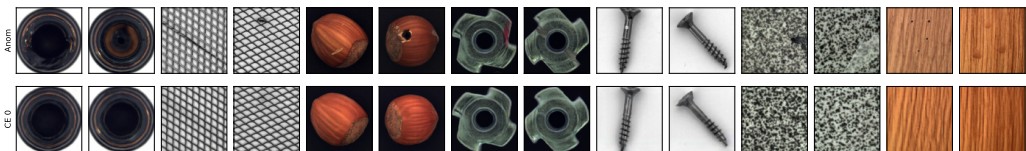

Figure 9: CEs for MVTec-AD and an anomaly detector trained with BCE and ImageNet-21k as OE. For each class, a different detector and CE generator was trained. The first row shows anomalies, the other corresponding CEs.

## D  METRICS

In this section, we provide details of the metrics used for the quantitative analysis in Section 4.3.

**Normality of counterfactuals**  To assess the normality of the generated CEs, we computed the AuROC of normal test samples against CEs generated for all ground-truth anomalies from the test set. The Area Under the ROC curve (AuROC) is a widely recognized metric in the AD literature

for comparing anomaly scores of normal and anomalous samples Hanley & McNeil (1982). An AuROC of 1 indicates perfect separation between anomalies and normal samples, $0.5$ corresponds to random guessing, and a score below $0.5$ suggests that anomalies appear more normal than the actual normal samples. To assess the normality of our CEs, we computed the AuROC with the anomalies being CEs. Then, an AuROC of significantly more than $0.5$ indicates that the CEs retain some degree of anomalousness according to the chosen detector. An AuROC of $0.5$ indicates that CEs appear completely normal, and for below $0.5$ the CEs are even more normal than the normal test samples. This may happen when the anomaly detector does not generalize perfectly and hence perceives some normal test samples as somewhat anomalous.

**Usefulness of counterfactuals for training AD**   To further assess the normality and realism of the CEs, we tested their ability to train a new anomaly detector. To this end, we replaced the entire normal training set with a collection of CEs generated for all ground-truth anomalies. With this modified training set, we retrained the AD methods, additionally using an outlier exposure set in case of BCE and HSC. If the CEs resemble normal images, the retrained anomaly detectors will outperform random guessing. We measure this by computing the AuROC for true normal vs. anomalous test samples and compare the outcome to the chance level, which is $0.5$.

**Realism of counterfactuals**   To assess the realism of generated samples, the standard approach involves computing the Fréchet inception distance (FID) introduced by Heusel et al. (2017) for GANs. The FID is the Wasserstein distance between the feature distributions of a generated dataset and a ground-truth dataset. The larger the distance, the less the generated dataset resembles the ground truth. The features are extracted using an InceptionNet v3 model Szegedy et al. (2015) trained on ImageNet. In this paper, we used the normal test set as ground truth and a collection of CEs for all test anomalies as the generated dataset. For a more intuitive scoring, we also computed a second FID with the test anomalies as the generated dataset. Then, we normalize the FID for CEs by dividing through the FID for test anomalies. The normalized FID is $100\%$ if the CEs are as realistic as the test anomalies, below $100\%$ if they are more realistic, and $0\%$ if they exactly match the normal test set. It is important to note that, although anomalies are naturally anomalous, they are still *realistic* in the sense that they come from the same classification dataset and thus follow the general distribution of, e.g., handwritten digits. A normalized FID of $100\%$ is therefore sufficient for a counterfactual to be expressive. A normalized FID of close to $0\%$ would actually be spurious, as the generator then seems to entirely reproduce normal samples that do not retain non-anomalous features from the anomaly.

**Disentanglement of counterfactuals**   We also evaluated the disentanglement of the sets of CEs for each anomaly. As introduced in Section 3, the proposed method includes a concept classifier trained to predict the concept of each CE. Consequently, we have a metric for assessing the disentanglement of the generated samples. The higher the accuracy of this classifier, the stronger the disentanglement of the generated CEs. We chose a rather small network for the concept classifier to encourage the network not to overfit on non-semantic features to predict the concepts.

## E  HYPERPARAMETERS

In this section, we provide an exhaustive enumeration of all the hyperparameters that we used for training our AD and CE module. All hyperparameters were adopted from existing research Ruff et al. (2018); Ghandeharioun et al. (2021); Liznerski et al. (2022). We start by describing the CE module, which is the same for all datasets and AD objectives. Then we separately describe the AD module and other hyperparameters for MNIST, Colored-MNIST, CIFAR-10, and GTSDB.

### E.1  THE CE MODULE

**Generator**   The generator is a wide ResNet Zagoruyko & Komodakis (2016) structured as an encoder-decoder network. The encoder consists of a sequential arrangement of a batch normalization layer, a convolutional layer with $64$ kernels, and three residual blocks. Each residual block comprises two sets, each containing a conditional batch normalization layer De Vries et al. (2017), followed by an activation function (ReLU), and a convolutional layer. The convolutional layers in these sets have $256$, $512$, and $1024$ kernels, respectively, for the first, second, and third block. The initial two residual blocks employ average pooling in each set to reduce the spatial dimension of the feature

maps by one-half of the input, while the third residual block is implemented without average pooling to maintain the spatial dimension. Conversely, the decoder follows a similar sequential arrangement, featuring three residual blocks, followed by a batch normalization layer, a final convolutional layer mapping to the image space, and an activation function (ReLU). Again, each residual block comprises two sets, each containing a conditional batch normalization layer, followed by RelU activation, and a convolutional layer. The convolutional layers in these sets have $1024$, $512$, and $256$ kernels, respectively, for the first, second, and third block. The first residual block in the decoder retains the spatial dimension, while the subsequent two residual blocks employ an interpolation layer in each set to upsample the spatial dimension by a multiplicative factor of 2 using nearest-neighbor interpolation. We apply spectral normalization to all layers of the decoder, following Miyato et al. (2018). The last layer of the decoder uses a tanh activation. The conditional information, i.e., the discretized target anomaly score $\alpha$ and the target concept $k$ are transformed into a single categorical condition and processed through the categorical conditional batch normalization layers.

**Discriminator**   The discriminator contains four residual blocks arranged sequentially, followed by a final linear layer mapping to a scalar. The first block is implemented with two convolutional layers with $64$ kernels, where the first layer is followed by a ReLU activation and the second layer is followed by an average pooling with a kernel size of 2. The next two residual blocks consist of two convolutional layers, where each one is preceded by a ReLU activation and followed by an average pooling layer in the end to halve the spatial dimension. The fourth residual block also contains two convolutional layers preceded by a ReLU, but does not use any downsampling. The number of kernels in the convolutional layers from the second to fourth block is $128$, $256$, and $512$, respectively. We apply spectral normalization to all layers.

**Concept Classifier**   The concept classifier is composed of two sequentially arranged residual blocks, succeeded by a linear layer with two outputs for the classification of two concepts. In the first residual block, three convolutional layers are employed with $64$ kernels each. The initial convolutional layer is succeeded by a ReLU activation, and the last two convolutional layers are followed by average pooling layers, which reduce the spatial dimension by a factor of two. The second residual block consists of two convolutional layers with $128$ kernels, each followed by a ReLU activation, followed by an average pooling with a kernel size of two. We take the sum over the remaining spatial dimension to prepare the output for the final linear layer. Again, we apply spectral normalization to all layers.

**Training**   We train the generator to generate CEs with two disentangled concepts and a discretized target anomaly score $\alpha \in 0, 0.5, 1$. The CE module is trained for $350$ ($2000$ for GTSDB) epochs with a batch size of $64$ normal and, if used, $64$ OE samples. The initial learning rate is set to $2e^{-4}$, with reductions by a multiplicative factor of $0.1$ occurring after $300$ and $325$ epochs. For GTSDB, we instead use an initial learning rate of $1e^{-4}$ and reduce it after $1750$ and $1900$ epochs. We employ the Adam optimizer, with the generator and discriminator optimized every 1 and 5 batches, respectively. The CE objective involves a combination of different losses which are weighted using $\lambda$ hyperparameters. Specifically, we set $\lambda_{gan} = 1$, $\lambda_{rec} = 100$, $\lambda_{\phi} = 1$, and $\lambda_r = 10$. For GTSDB, we instead set $\lambda_{gan} = 5$, $\lambda_{rec} = 20$, $\lambda_{\phi} = 1$, and $\lambda_r = 10$. For INN, we use a different set of hyperparameters. We set $\lambda_{gan} = 10$, $\lambda_{rec} = 1$, $\lambda_{\phi} = 1$, and $\lambda_r = 0.5$. Also, we consider only $\alpha = 0$, as we train the generator with only OE samples to reduce the training time, while the discriminator is trained with normal and generated samples. Due to the immense VRAM requirements of the diffusion model, we train with a batch size of 1 and use the running statistics of all BatchNorm layers during training. The initial learning rate is set to $1e^{-4}$. It is reduced by a factor of $0.5$ at $100, 120, 130, 140$, and $145$ epochs. The model is trained for $150$ epochs in total.

### E.2   AD ON MNIST

For MNIST and all the following datasets, we trained anomaly detectors with a binary cross entropy (BCE) and hypersphere classification (HSC) loss, both with Outlier Exposure (OE) Hendrycks et al. (2019a), as well as DSVDD Ruff et al. (2018) without OE.

We use a LeNet-style neural network comprising layers arranged sequentially without residual connections. The network contains four convolutional layers and two fully-connected layers. Each convolutional layer is followed by batch normalization, a leaky ReLU activation, and max-pooling. The first fully connected layer is followed by batch normalization and a leaky ReLU activation, while

the last layer is only a linear transformation. The number of kernels in the convolutional layers is, from first to last, 4, 8, 16, and 32. The kernel size is increased from the default of 3 to 5 for all of these. The two fully connected layers have 64 and 32 units, respectively. For DSVDD we remove bias from the network, following Ruff et al. (2018), and for BCE we add another linear layer with sigmoid activation.

We used Adam for optimization and balanced every batch to contain 128 normal and 128 OE samples during training. We trained the AD model for 80 epochs starting with a learning rate of $1e^{-4}$, which we reduced to $1e^{-5}$ after 60 epochs.

### E.3 AD ON COLORED-MNIST

Based on the MNIST dataset, we create Colored-MNIST where for each sample in MNIST six copies in different colors (red, yellow, green, cyan, blue, pink) are created. We use a colored version of EMNIST as OE. The network for Colored-MNIST is a slight variation of the AD network used on MNIST. We remove the last convolutional layer and change the number of kernels for the convolutional layers to 16, 32, and 64, respectively.

We use Adam for optimization, balance every batch to contain 128 normal and 128 OE samples during training, and train the AD model for 120 epochs, starting with a learning rate of $5e^{-5}$, reduced to $5e^{-6}$ after 100 epochs.

### E.4 AD ON CIFAR-10

For CIFAR-10, previous work used 80 Mio. Tiny Images as OE Hendrycks et al. (2019b). However, since 80 Mio. Tiny Images has officially been withdrawn due to offensive data, we instead use the disjunct CIFAR-100 dataset as OE. We found that this does not cause a significant drop of performance. Again, we use a slight variation of the AD network used on MNIST. We remove the last convolutional layer and change the number of kernels for the convolutional layers to 32, 64, and 128, respectively. The fully connected layers have 512 and 256 units instead.

We use Adam for optimization and balance every batch to contain 128 normal and 128 OE samples during training. We train the AD model for 200 epochs starting with a learning rate of $1e^{-3}$, which we reduce by a factor of 0.1 after 100 and 150 epochs.

### E.5 AD ON GTSDB

We use the same setup on GTSDB as on CIFAR-10.

### E.6 AD ON IMAGENET-NEIGHBORS

For ImageNet-Neighbors (INN), we use the disjoint ImageNet-21k as OE and the same WideResNet architecture as in (Hendrycks et al., 2019b; Liznerski et al., 2022). We use Adam for optimization and balance every batch to contain 64 normal and 64 OE samples during training. We train the AD model for 150 epochs starting with a learning rate of $1e^{-3}$, which we reduce by a factor of 0.1 after 100 and 125 epochs.

## F COMPUTE RESOURCES

Most of the experiments with MNIST, Colored-MNIST, CIFAR-10, and GTSDB were carried out on a NVIDIA DGX-1 server containing 8 GV100 GPUs with 32 GB memory. For Colored-MNIST, each experiment with one seed and normal class definition took around one and a half days. For MNIST and CIFAR-10, each experiment took approximately 8 hours. Each GTSDB experiment took only about 3 hours. The time to run each experiment varies depending on the precise setup. For the INN experiments, most experiments were carried out on a NVIDIA DGX A-100 server with 8 A100 GPUs with 40 GB memory. One experiment with one seed and normal class definition took approximately 10 days.

## G    FULL QUANTITATIVE RESULTS PER NORMAL CLASS

In the main paper, we proposed a method to generate counterfactual explanations (CEs) for deep anomaly detection on images. We also presented several objective evaluation techniques to validate their performance on MNIST, Colored-MNIST (C-MNIST), CIFAR-10, GTSDB, and ImageNet-Neighbors (INN) across different definitions of normality. Following previous work on semantic image-AD Ruff et al. (2018); Golan & El-Yaniv (2018); Hendrycks et al. (2019a;b); Ruff et al. (2020); Tack et al. (2020); Ruff et al. (2021); Liznerski et al. (2021; 2022), we turned classification datasets into AD benchmarks by defining a subset of the classes to be normal and using the remainder as ground-truth anomalies for testing. If only one class is normal, this approach is termed *one vs. rest* AD. Apart from investigating one vs. rest, we also explored a variation with multiple classes being normal. For our experiments, we considered all classes of MNIST and CIFAR-10 as single normal classes and, to keep the computational load at a reasonable level, a subset of 20 normal class combinations. The class combinations were chosen from $\{(i, (i+1) \mod 10) \, | \, i \in \{0, \ldots, 9\}\} \cup \{(i, (i+2) \mod 10) \, | \, i \in \{0, \ldots, 9\}\}$. For Colored-MNIST, we considered all combinations of color and the digit one as normal. For GTSDB, we considered the following pairs of street signs as normal: all four combinations of speed limit signs, the "give way" and stop sign, and the "danger" and "construction" warning sign. Additionally, we considered four larger sets of normal classes: all "restriction ends" signs, all speed limit signs, all blue signs, and all warning signs. In total, we consider ten different scenarios of normal definitions for GTSDB.

We introduced ImageNet-Neighbors (INN), which is a subset of ImageNet-1K. As before, we define an AD setup by considering one of the classes normal. However, instead of using the entire remainder as ground-truth test anomalies, we choose only the ten most similar classes, based on the Wu-Palmer similarity metric (Wu & Palmer, 1994), as test anomalies. This AD setup becomes harder as compared to the usual one vs. rest AD setup (Hendrycks et al., 2019a), as the anomalies are more similar to the normal class and thus harder to detect, especially in an unsupervised manner. In this paper, we consider five different AD setups for INN. (1) An airliner is normal with airship, wreck, warplane, balloon, monocycle, fireboat, schooner, space shuttle, pirate ship, and gondola as test anomalies. (2) An ambulance is normal with limousine, taxi, waggon, racing car, minivan, jeep, sports car, golf cart, Model T, and convertible as test anomalies. (3) A black widow (spider) is normal with centipede, trilobite, wolf spider, garden spider, barn spider, harvestman, scorpion, black and gold garden spider, tarantula, and tick as test anomalies. (4) A lion is normal with cougar, cheetah, jaguar, tiger cat, leopard, snow leopard, lynx, tiger, tabby cat, and Siamese cat as test anomalies. (5) A zebra is normal with sorrel, llama, warthog, boar, hamster, armadillo, hog, beaver, Arabian camel, and hippo as test anomalies.

For each scenario on each dataset, a new AD model and counterfactual generator was trained for four random seeds. Due to space constraints, we reported our quantitative results averaged over all normal definitions in the main paper. Here, we report results averaged over four random seeds separately for each normal definition. We consider the following metrics from the main paper:

- The AD AuROC (Section 4.3.2) is the AuROC of normal vs. anomalous test samples, thereby measuring the AD performance of the AD model. $50\%$ is random, $100\%$ indicates optimal separation.

- The CF AuROC (Section 4.3.1) is the AuROC of normal test samples vs. counterfactuals. The counterfactuals appear entirely normal for an AuROC $\leq 50\%$.

- The Sub. AuROC (Section 4.3.2) is the AuROC of normal vs. anomalous test samples when the AD is trained with counterfactuals in place of the normal training set.

- The $FID_N$ (Section 4.3.3) denotes the normalized FID scores. $0\%$ indicates that the counterfactuals follow the same feature distribution as normal samples, $100\%$ as anomalies, which are also realistic, and above $100\%$ indicates less realistic counterfactuals.

- The Concept Acc (Section 4.3.4) is the accuracy of the concept classifier. A $100\%$ accuracy indicates optimal disentanglement of the concepts.

Additionally, we report the "Score distance", which is the L1 distance between the average anomaly score of normal and anomalous test samples. Note that the L1 distance between normal training data and OE samples is usually 1. Thus, the "Score distance" measures the generalizability of the AD model to ground-truth anomalies in terms of anomaly score calibration.

Tables 5, 6, and 7 show results for MNIST and single normal classes for BCE, HSC, and DSVDD, respectively. In Tables 8, 9, and 10, we instead report results for CIFAR-10 and single normal classes for BCE, HSC, and DSVDD, respectively. Tables 11, 12, and 13 show results for Colored-MNIST (here abbreviated as C-MNIST) for BCE, HSC, and DSVDD, respectively. Tables 14, 15, and 16 show results for GTSDB and combined normal classes for BCE, HSC, and DSVDD, respectively. Tables 17, 18, and 19 show results for MNIST and combined normal classes for BCE, HSC, and DSVDD, respectively. Tables 20, 21, and 22 show results for CIFAR-10 and combined normal classes for BCE, HSC, and DSVDD, respectively. Tables 23 and 24 show results for ImageNet-Neighbors and single normal classes for BCE and HSC, respectively.

Table 5: AD and explanation performance averaged over 4 random seeds on MNIST for BCE (OE). Each row shows results for a different normal definition.

| | AD | | Explanation | | | |
| Normal | AuROC | Score distance | CF AuROC | Sub. AuROC | $FID_N$ | Concept Acc |
|---|---|---|---|---|---|---|
| zero | 0.99 ± 0.0010 | 0.78 ± 0.0079 | 0.76 ± 0.0684 | 0.93 ± 0.0104 | 0.42 ± 0.0366 | 0.97 ± 0.0360 |
| one | 1.00 ± 0.0005 | 0.87 ± 0.0155 | 0.66 ± 0.0977 | 0.97 ± 0.0107 | 0.47 ± 0.4474 | 0.99 ± 0.0082 |
| two | 0.97 ± 0.0083 | 0.69 ± 0.0379 | 0.75 ± 0.0253 | 0.85 ± 0.0183 | 0.56 ± 0.0431 | 0.87 ± 0.0505 |
| three | 0.99 ± 0.0018 | 0.67 ± 0.0286 | 0.77 ± 0.0242 | 0.94 ± 0.0073 | 0.33 ± 0.0392 | 0.89 ± 0.0834 |
| four | 0.97 ± 0.0090 | 0.75 ± 0.0359 | 0.70 ± 0.0787 | 0.88 ± 0.0457 | 0.48 ± 0.0954 | 0.91 ± 0.0563 |
| five | 0.97 ± 0.0058 | 0.65 ± 0.0398 | 0.66 ± 0.0076 | 0.84 ± 0.0184 | 0.44 ± 0.0405 | 0.98 ± 0.0252 |
| six | 1.00 ± 0.0010 | 0.90 ± 0.0106 | 0.71 ± 0.0527 | 0.98 ± 0.0066 | 0.33 ± 0.0348 | 0.96 ± 0.0359 |
| seven | 0.96 ± 0.0107 | 0.71 ± 0.0275 | 0.70 ± 0.0519 | 0.92 ± 0.0133 | 0.50 ± 0.0464 | 0.96 ± 0.0281 |
| eight | 0.95 ± 0.0102 | 0.54 ± 0.0337 | 0.72 ± 0.0817 | 0.87 ± 0.0054 | 0.31 ± 0.0271 | 0.94 ± 0.0794 |
| nine | 0.96 ± 0.0092 | 0.60 ± 0.0329 | 0.77 ± 0.0147 | 0.94 ± 0.0080 | 0.47 ± 0.0593 | 0.97 ± 0.0189 |
| mean | 0.98 ± 0.0154 | 0.72 ± 0.1067 | 0.72 ± 0.0400 | 0.91 ± 0.0456 | 0.43 ± 0.0808 | 0.94 ± 0.0385 |

Table 6: AD and explanation performance averaged over 4 random seeds on MNIST for HSC (OE). Each row shows results for a different normal definition.

| | AD | | Explanation | | | |
| Normal | AuROC | Score distance | CF AuROC | Sub. AuROC | $FID_N$ | Concept Acc |
|---|---|---|---|---|---|---|
| zero | 0.99 ± 0.0011 | 0.81 ± 0.0306 | 0.84 ± 0.0772 | 0.91 ± 0.0101 | 0.58 ± 0.1412 | 0.98 ± 0.0106 |
| one | 1.00 ± 0.0011 | 0.89 ± 0.0231 | 0.88 ± 0.0783 | 0.95 ± 0.0089 | 0.60 ± 0.3820 | 0.90 ± 0.0868 |
| two | 0.98 ± 0.0013 | 0.72 ± 0.0338 | 0.77 ± 0.0332 | 0.77 ± 0.0438 | 0.80 ± 0.3295 | 0.92 ± 0.0575 |
| three | 0.98 ± 0.0056 | 0.67 ± 0.0166 | 0.82 ± 0.0717 | 0.85 ± 0.0209 | 0.48 ± 0.2057 | 0.83 ± 0.1941 |
| four | 0.96 ± 0.0038 | 0.73 ± 0.0269 | 0.80 ± 0.0658 | 0.84 ± 0.0394 | 0.83 ± 0.2911 | 0.81 ± 0.1526 |
| five | 0.96 ± 0.0054 | 0.62 ± 0.0334 | 0.83 ± 0.0603 | 0.70 ± 0.1316 | 0.77 ± 0.1088 | 0.92 ± 0.1010 |
| six | 1.00 ± 0.0010 | 0.88 ± 0.0211 | 0.77 ± 0.0607 | 0.98 ± 0.0076 | 0.84 ± 0.3493 | 0.95 ± 0.0547 |
| seven | 0.97 ± 0.0052 | 0.71 ± 0.0066 | 0.70 ± 0.0319 | 0.92 ± 0.0112 | 0.52 ± 0.0301 | 0.91 ± 0.0675 |
| eight | 0.95 ± 0.0069 | 0.52 ± 0.0334 | 0.89 ± 0.0278 | 0.73 ± 0.0590 | 0.88 ± 0.3052 | 0.94 ± 0.0739 |
| nine | 0.97 ± 0.0043 | 0.59 ± 0.0192 | 0.80 ± 0.0227 | 0.92 ± 0.0031 | 0.53 ± 0.0739 | 0.91 ± 0.0512 |
| mean | 0.98 ± 0.0157 | 0.72 ± 0.1156 | 0.81 ± 0.0526 | 0.86 ± 0.0919 | 0.68 ± 0.1464 | 0.91 ± 0.0478 |

Table 7: AD and explanation performance averaged over 4 random seeds on MNIST for DSVDD. Each row shows results for a different normal definition.

| Normal | AD | | Explanation | | | |
|---|---|---|---|---|---|---|
| | AuROC | Score distance | CF AuROC | Sub. AuROC | $FID_N$ | Concept Acc |
| zero | $0.82 \pm 0.0685$ | $0.01 \pm 0.0038$ | $0.76 \pm 0.0870$ | $0.41 \pm 0.0680$ | $1.16 \pm 0.5100$ | $0.96 \pm 0.0467$ |
| one | $1.00 \pm 0.0020$ | $0.05 \pm 0.0086$ | $0.99 \pm 0.0054$ | $0.76 \pm 0.1219$ | $1.02 \pm 0.0600$ | $0.84 \pm 0.1254$ |
| two | $0.72 \pm 0.1254$ | $0.01 \pm 0.0057$ | $0.69 \pm 0.1664$ | $0.34 \pm 0.0203$ | $0.89 \pm 0.0117$ | $0.49 \pm 0.1150$ |
| three | $0.72 \pm 0.0274$ | $0.00 \pm 0.0036$ | $0.70 \pm 0.0545$ | $0.42 \pm 0.0527$ | $0.90 \pm 0.0234$ | $0.59 \pm 0.1276$ |
| four | $0.72 \pm 0.0517$ | $0.01 \pm 0.0040$ | $0.65 \pm 0.0669$ | $0.46 \pm 0.0180$ | $0.88 \pm 0.1156$ | $0.80 \pm 0.1840$ |
| five | $0.73 \pm 0.0316$ | $0.01 \pm 0.0050$ | $0.71 \pm 0.0562$ | $0.44 \pm 0.0632$ | $0.97 \pm 0.0869$ | $0.87 \pm 0.1221$ |
| six | $0.83 \pm 0.0964$ | $0.01 \pm 0.0126$ | $0.80 \pm 0.1238$ | $0.44 \pm 0.0466$ | $1.08 \pm 0.0339$ | $0.84 \pm 0.1877$ |
| seven | $0.84 \pm 0.0450$ | $0.01 \pm 0.0135$ | $0.80 \pm 0.0533$ | $0.46 \pm 0.0858$ | $1.04 \pm 0.0408$ | $0.88 \pm 0.0291$ |
| eight | $0.70 \pm 0.0359$ | $0.00 \pm 0.0007$ | $0.69 \pm 0.0440$ | $0.46 \pm 0.0792$ | $0.99 \pm 0.0775$ | $0.82 \pm 0.0962$ |
| nine | $0.81 \pm 0.0331$ | $0.01 \pm 0.0056$ | $0.74 \pm 0.0568$ | $0.44 \pm 0.0599$ | $1.09 \pm 0.0822$ | $0.65 \pm 0.3127$ |
| mean | $0.79 \pm 0.0865$ | $0.01 \pm 0.0119$ | $0.75 \pm 0.0916$ | $0.46 \pm 0.1050$ | $1.00 \pm 0.0876$ | $0.78 \pm 0.1410$ |

Table 8: AD and explanation performance averaged over 4 random seeds on CIFAR-10 for BCE OE. Each row shows results for a different normal definition.

| Normal | AD | | Explanation | | | |
| | AuROC | Score distance | CF AuROC | Sub. AuROC | $FID_N$ | Concept Acc |
|---|---|---|---|---|---|---|
| airplane | 0.96 ± 0.0009 | 0.78 ± 0.0083 | 0.47 ± 0.0372 | 0.65 ± 0.0322 | 1.48 ± 0.1439 | 0.93 ± 0.0659 |
| automobile | 0.99 ± 0.0005 | 0.87 ± 0.0026 | 0.62 ± 0.0540 | 0.62 ± 0.0347 | 1.08 ± 0.0582 | 0.92 ± 0.0757 |
| bird | 0.93 ± 0.0030 | 0.65 ± 0.0020 | 0.42 ± 0.0378 | 0.53 ± 0.0138 | 1.42 ± 0.0777 | 0.99 ± 0.0069 |
| cat | 0.91 ± 0.0035 | 0.55 ± 0.0127 | 0.30 ± 0.0054 | 0.53 ± 0.0159 | 1.37 ± 0.0773 | 0.91 ± 0.1449 |
| deer | 0.96 ± 0.0020 | 0.74 ± 0.0043 | 0.40 ± 0.0209 | 0.53 ± 0.0103 | 1.09 ± 0.1095 | 0.99 ± 0.0151 |
| dog | 0.94 ± 0.0013 | 0.64 ± 0.0051 | 0.36 ± 0.0061 | 0.57 ± 0.0134 | 1.23 ± 0.0777 | 0.93 ± 0.1008 |
| frog | 0.98 ± 0.0011 | 0.79 ± 0.0067 | 0.50 ± 0.0247 | 0.54 ± 0.0127 | 0.80 ± 0.0652 | 0.88 ± 0.1341 |
| horse | 0.98 ± 0.0006 | 0.82 ± 0.0060 | 0.59 ± 0.0303 | 0.64 ± 0.0213 | 1.21 ± 0.1013 | 0.99 ± 0.0107 |
| ship | 0.98 ± 0.0002 | 0.85 ± 0.0032 | 0.55 ± 0.0098 | 0.72 ± 0.0300 | 0.93 ± 0.0810 | 0.89 ± 0.0760 |
| truck | 0.97 ± 0.0018 | 0.78 ± 0.0080 | 0.54 ± 0.0602 | 0.56 ± 0.0242 | 1.03 ± 0.1231 | 0.88 ± 0.2031 |
| mean | 0.96 ± 0.0252 | 0.75 ± 0.0964 | 0.47 ± 0.1000 | 0.59 ± 0.0610 | 1.16 ± 0.2078 | 0.93 ± 0.0429 |

Table 9: AD and explanation performance averaged over 4 random seeds on CIFAR-10 for HSC OE. Each row shows results for a different normal definition.

| Normal | AD | | Explanation | | | |
| | AuROC | Score distance | CF AuROC | Sub. AuROC | $FID_N$ | Concept Acc |
|---|---|---|---|---|---|---|
| airplane | 0.96 ± 0.0012 | 0.75 ± 0.0056 | 0.51 ± 0.0754 | 0.52 ± 0.0111 | 2.95 ± 0.1509 | 0.89 ± 0.0873 |
| automobile | 0.99 ± 0.0005 | 0.85 ± 0.0030 | 0.58 ± 0.0152 | 0.59 ± 0.0129 | 1.71 ± 0.1914 | 0.99 ± 0.0054 |
| bird | 0.93 ± 0.0015 | 0.62 ± 0.0018 | 0.46 ± 0.0293 | 0.52 ± 0.0149 | 4.81 ± 0.2365 | 1.00 ± 0.0007 |
| cat | 0.90 ± 0.0020 | 0.53 ± 0.0072 | 0.43 ± 0.0255 | 0.52 ± 0.0088 | 3.98 ± 0.4753 | 1.00 ± 0.0009 |
| deer | 0.96 ± 0.0007 | 0.71 ± 0.0040 | 0.51 ± 0.0121 | 0.57 ± 0.0230 | 3.45 ± 0.3143 | 1.00 ± 0.0000 |
| dog | 0.95 ± 0.0012 | 0.65 ± 0.0047 | 0.46 ± 0.0317 | 0.53 ± 0.0257 | 3.09 ± 0.2897 | 1.00 ± 0.0023 |
| frog | 0.98 ± 0.0004 | 0.77 ± 0.0043 | 0.52 ± 0.0062 | 0.57 ± 0.0569 | 2.92 ± 0.4138 | 1.00 ± 0.0009 |
| horse | 0.98 ± 0.0008 | 0.79 ± 0.0040 | 0.54 ± 0.0466 | 0.54 ± 0.0281 | 3.13 ± 0.0463 | 1.00 ± 0.0001 |
| ship | 0.98 ± 0.0003 | 0.83 ± 0.0027 | 0.48 ± 0.0257 | 0.56 ± 0.0316 | 1.86 ± 0.5187 | 1.00 ± 0.0032 |
| truck | 0.97 ± 0.0011 | 0.77 ± 0.0055 | 0.51 ± 0.0257 | 0.57 ± 0.0623 | 2.19 ± 0.1318 | 1.00 ± 0.0010 |
| mean | 0.96 ± 0.0254 | 0.73 ± 0.0939 | 0.50 ± 0.0438 | 0.55 ± 0.0259 | 3.01 ± 0.8998 | 0.99 ± 0.0325 |

Table 10: AD and explanation performance averaged over 4 random seeds on CIFAR-10 for DSVDD. Each row shows results for a different normal definition.

| Normal | AD | | Explanation | | | |
| | AuROC | Score distance | CF AuROC | Sub. AuROC | $FID_N$ | Concept Acc |
|---|---|---|---|---|---|---|
| airplane | 0.48 ± 0.0952 | -0.00 ± 0.0022 | 0.54 ± 0.0733 | 0.45 ± 0.0265 | 1.28 ± 0.0382 | 0.98 ± 0.0114 |
| automobile | 0.51 ± 0.0339 | 0.00 ± 0.0003 | 0.52 ± 0.0606 | 0.49 ± 0.0198 | 1.15 ± 0.0266 | 0.99 ± 0.0076 |
| bird | 0.54 ± 0.0375 | 0.00 ± 0.0005 | 0.52 ± 0.0601 | 0.51 ± 0.0133 | 1.23 ± 0.0548 | 0.91 ± 0.1548 |
| cat | 0.52 ± 0.0216 | 0.00 ± 0.0008 | 0.51 ± 0.0513 | 0.50 ± 0.0260 | 1.38 ± 0.1380 | 0.98 ± 0.0221 |
| deer | 0.65 ± 0.0312 | 0.01 ± 0.0030 | 0.62 ± 0.0996 | 0.53 ± 0.0611 | 1.12 ± 0.0467 | 1.00 ± 0.0028 |
| dog | 0.53 ± 0.0259 | 0.00 ± 0.0030 | 0.51 ± 0.0296 | 0.50 ± 0.0195 | 1.21 ± 0.0830 | 0.96 ± 0.0523 |
| frog | 0.60 ± 0.0692 | 0.01 ± 0.0027 | 0.54 ± 0.0371 | 0.57 ± 0.0747 | 0.99 ± 0.0550 | 0.99 ± 0.0074 |
| horse | 0.56 ± 0.0253 | 0.00 ± 0.0025 | 0.53 ± 0.0281 | 0.51 ± 0.0143 | 1.21 ± 0.0094 | 1.00 ± 0.0037 |
| ship | 0.57 ± 0.0543 | 0.00 ± 0.0010 | 0.58 ± 0.0350 | 0.53 ± 0.0561 | 0.97 ± 0.0611 | 0.93 ± 0.0758 |
| truck | 0.58 ± 0.0673 | 0.00 ± 0.0008 | 0.58 ± 0.0470 | 0.48 ± 0.0224 | 1.10 ± 0.0258 | 0.97 ± 0.0417 |
| mean | 0.55 ± 0.0473 | 0.00 ± 0.0022 | 0.55 ± 0.0336 | 0.51 ± 0.0315 | 1.16 ± 0.1195 | 0.97 ± 0.0287 |

Table 11: AD and explanation performance averaged over 4 random seeds on C-MNIST for BCE (OE). Each row shows results for a different normal definition.

| Normal | AD | | Explanation | | | |
| | AuROC | Score distance | CF AuROC | Sub. AuROC | FID$_N$ | Concept Acc |
|---|---|---|---|---|---|---|
| gray+one | $0.96 \pm 0.0037$ | $0.17 \pm 0.0127$ | $0.55 \pm 0.1105$ | $0.75 \pm 0.0429$ | $0.75 \pm 0.3352$ | $0.96 \pm 0.0327$ |
| yellow+one | $0.97 \pm 0.0027$ | $0.24 \pm 0.0129$ | $0.56 \pm 0.0252$ | $0.74 \pm 0.0082$ | $0.60 \pm 0.1572$ | $1.00 \pm 0.0001$ |
| cyan+one | $0.96 \pm 0.0138$ | $0.19 \pm 0.0373$ | $0.54 \pm 0.0410$ | $0.83 \pm 0.0180$ | $0.38 \pm 0.0340$ | $1.00 \pm 0.0007$ |
| green+one | $0.99 \pm 0.0044$ | $0.49 \pm 0.0546$ | $0.58 \pm 0.0457$ | $0.80 \pm 0.0676$ | $0.60 \pm 0.2606$ | $1.00 \pm 0.0001$ |
| blue+one | $0.98 \pm 0.0034$ | $0.48 \pm 0.0110$ | $0.55 \pm 0.0075$ | $0.81 \pm 0.0640$ | $0.52 \pm 0.1925$ | $1.00 \pm 0.0002$ |
| pink+one | $0.97 \pm 0.0021$ | $0.25 \pm 0.0193$ | $0.57 \pm 0.0279$ | $0.88 \pm 0.0127$ | $0.43 \pm 0.0647$ | $1.00 \pm 0.0003$ |
| red+one | $0.98 \pm 0.0031$ | $0.42 \pm 0.0364$ | $0.54 \pm 0.1100$ | $0.83 \pm 0.0938$ | $0.69 \pm 0.4817$ | $1.00 \pm 0.0015$ |
| mean | $0.97 \pm 0.0101$ | $0.32 \pm 0.1265$ | $0.56 \pm 0.0154$ | $0.81 \pm 0.0451$ | $0.57 \pm 0.1240$ | $0.99 \pm 0.0132$ |

Table 12: AD and explanation performance averaged over 4 random seeds on C-MNIST for HSC (OE). Each row shows results for a different normal definition.

| Normal | AD | | Explanation | | | |
| | AuROC | Score distance | CF AuROC | Sub. AuROC | FID$_N$ | Concept Acc |
|---|---|---|---|---|---|---|
| gray+one | $0.92 \pm 0.0075$ | $0.27 \pm 0.0410$ | $0.51 \pm 0.0486$ | $0.76 \pm 0.0457$ | $0.86 \pm 0.1567$ | $0.99 \pm 0.0136$ |
| yellow+one | $0.94 \pm 0.0251$ | $0.43 \pm 0.0509$ | $0.54 \pm 0.0615$ | $0.82 \pm 0.0081$ | $0.82 \pm 0.2713$ | $1.00 \pm 0.0020$ |
| cyan+one | $0.97 \pm 0.0196$ | $0.39 \pm 0.0630$ | $0.56 \pm 0.0296$ | $0.88 \pm 0.0462$ | $0.63 \pm 0.2201$ | $1.00 \pm 0.0000$ |
| green+one | $0.98 \pm 0.0139$ | $0.52 \pm 0.0258$ | $0.56 \pm 0.0323$ | $0.89 \pm 0.0102$ | $0.94 \pm 0.2280$ | $1.00 \pm 0.0005$ |
| blue+one | $0.99 \pm 0.0028$ | $0.65 \pm 0.0159$ | $0.66 \pm 0.0896$ | $0.75 \pm 0.1384$ | $1.66 \pm 1.1219$ | $0.94 \pm 0.0834$ |
| pink+one | $0.94 \pm 0.0139$ | $0.38 \pm 0.0323$ | $0.52 \pm 0.0751$ | $0.83 \pm 0.0339$ | $0.83 \pm 0.0292$ | $1.00 \pm 0.0015$ |
| red+one | $0.98 \pm 0.0031$ | $0.60 \pm 0.0127$ | $0.57 \pm 0.0244$ | $0.78 \pm 0.0674$ | $0.93 \pm 0.3331$ | $1.00 \pm 0.0055$ |
| mean | $0.96 \pm 0.0231$ | $0.46 \pm 0.1226$ | $0.56 \pm 0.0472$ | $0.82 \pm 0.0482$ | $0.95 \pm 0.3047$ | $0.99 \pm 0.0198$ |

Table 13: AD and explanation performance averaged over 4 random seeds on C-MNIST for DSVDD. Each row shows results for a different normal definition.

| Normal | AD | | Explanation | | | |
| | AuROC | Score distance | CF AuROC | Sub. AuROC | FID$_N$ | Concept Acc |
|---|---|---|---|---|---|---|
| gray+one | $0.73 \pm 0.0350$ | $0.00 \pm 0.0001$ | $0.56 \pm 0.0449$ | $0.71 \pm 0.0755$ | $0.85 \pm 0.2079$ | $0.91 \pm 0.0834$ |
| yellow+one | $0.86 \pm 0.0262$ | $0.00 \pm 0.0010$ | $0.60 \pm 0.0595$ | $0.65 \pm 0.0639$ | $0.82 \pm 0.2240$ | $1.00 \pm 0.0044$ |
| cyan+one | $0.83 \pm 0.0866$ | $0.00 \pm 0.0005$ | $0.61 \pm 0.0781$ | $0.63 \pm 0.0589$ | $0.79 \pm 0.0524$ | $0.99 \pm 0.0057$ |
| green+one | $0.64 \pm 0.1336$ | $0.00 \pm 0.0003$ | $0.57 \pm 0.0250$ | $0.60 \pm 0.0755$ | $0.69 \pm 0.0350$ | $1.00 \pm 0.0019$ |
| blue+one | $0.78 \pm 0.1502$ | $0.00 \pm 0.0001$ | $0.68 \pm 0.2173$ | $0.42 \pm 0.1223$ | $1.01 \pm 0.1866$ | $1.00 \pm 0.0016$ |
| pink+one | $0.75 \pm 0.1343$ | $0.00 \pm 0.0001$ | $0.67 \pm 0.1040$ | $0.61 \pm 0.0999$ | $0.85 \pm 0.0998$ | $0.97 \pm 0.0214$ |
| red+one | $0.79 \pm 0.0424$ | $0.00 \pm 0.0004$ | $0.62 \pm 0.0917$ | $0.57 \pm 0.1607$ | $0.81 \pm 0.1763$ | $0.99 \pm 0.0149$ |
| mean | $0.77 \pm 0.0650$ | $0.00 \pm 0.0003$ | $0.61 \pm 0.0430$ | $0.60 \pm 0.0841$ | $0.83 \pm 0.0875$ | $0.98 \pm 0.0297$ |

Table 14: AD and explanation performance averaged over 4 random seeds on GTSDB for BCE OE. Each row shows results for a different normal definition.

| Normal | AD | | Explanation | | | |
| | AuROC | Score distance | CF AuROC | Sub. AuROC | $FID_N$ | Concept Acc |
| --- | --- | --- | --- | --- | --- | --- |
| speed limit 30 + 50 | $0.92 \pm 0.0037$ | $0.65 \pm 0.0103$ | $0.51 \pm 0.0563$ | $0.88 \pm 0.0158$ | $0.77 \pm 0.3590$ | $1.00 \pm 0.0018$ |
| speed limit 50 + 70 | $0.88 \pm 0.0151$ | $0.59 \pm 0.0188$ | $0.49 \pm 0.0576$ | $0.86 \pm 0.0066$ | $0.69 \pm 0.3249$ | $0.99 \pm 0.0080$ |
| speed limit 70 + 100 | $0.88 \pm 0.0053$ | $0.57 \pm 0.0048$ | $0.55 \pm 0.0708$ | $0.89 \pm 0.0136$ | $0.42 \pm 0.1348$ | $0.99 \pm 0.0130$ |
| speed limit 100 + 120 | $0.89 \pm 0.0200$ | $0.55 \pm 0.0409$ | $0.49 \pm 0.1331$ | $0.87 \pm 0.0297$ | $0.51 \pm 0.0854$ | $0.99 \pm 0.0115$ |
| give way + stop | $0.99 \pm 0.0021$ | $0.89 \pm 0.0131$ | $0.66 \pm 0.0758$ | $0.81 \pm 0.1369$ | $2.29 \pm 0.4255$ | $0.99 \pm 0.0184$ |
| danger + construction warning | $0.93 \pm 0.0078$ | $0.73 \pm 0.0072$ | $0.43 \pm 0.0799$ | $0.91 \pm 0.0155$ | $3.60 \pm 0.5202$ | $1.00 \pm 0.0040$ |
| all restriction ends signs | $1.00 \pm 0.0029$ | $0.90 \pm 0.0167$ | $0.56 \pm 0.1341$ | $1.00 \pm 0.0033$ | $0.24 \pm 0.1129$ | $0.97 \pm 0.0183$ |
| all speed limit signs | $0.99 \pm 0.0016$ | $0.79 \pm 0.0226$ | $0.54 \pm 0.0172$ | $0.96 \pm 0.0085$ | $0.41 \pm 0.0870$ | $0.99 \pm 0.0134$ |
| all blue signs | $1.00 \pm 0.0023$ | $0.93 \pm 0.0131$ | $0.40 \pm 0.0381$ | $0.90 \pm 0.0258$ | $0.64 \pm 0.1553$ | $0.98 \pm 0.0109$ |
| all warning signs | $0.96 \pm 0.0089$ | $0.89 \pm 0.0132$ | $0.38 \pm 0.0343$ | $0.95 \pm 0.0035$ | $1.51 \pm 0.5426$ | $0.99 \pm 0.0076$ |
| mean | $0.94 \pm 0.0474$ | $0.75 \pm 0.1437$ | $0.50 \pm 0.0803$ | $0.90 \pm 0.0526$ | $1.11 \pm 1.0182$ | $0.99 \pm 0.0085$ |

Table 15: AD and explanation performance averaged over 4 random seeds on GTSDB for HSC OE. Each row shows results for a different normal definition.

| Normal | AD | | Explanation | | | |
| | AuROC | Score distance | CF AuROC | Sub. AuROC | $FID_N$ | Concept Acc |
| --- | --- | --- | --- | --- | --- | --- |
| speed limit 30 + 50 | $0.88 \pm 0.0014$ | $0.63 \pm 0.0126$ | $0.31 \pm 0.1032$ | $0.88 \pm 0.0113$ | $0.79 \pm 0.2196$ | $0.96 \pm 0.0420$ |
| speed limit 50 + 70 | $0.89 \pm 0.0111$ | $0.57 \pm 0.0170$ | $0.49 \pm 0.1537$ | $0.85 \pm 0.0135$ | $1.45 \pm 0.6565$ | $1.00 \pm 0.0000$ |
| speed limit 70 + 100 | $0.86 \pm 0.0164$ | $0.56 \pm 0.0146$ | $0.60 \pm 0.1389$ | $0.85 \pm 0.0379$ | $0.69 \pm 0.4033$ | $0.91 \pm 0.0807$ |
| speed limit 100 + 120 | $0.85 \pm 0.0112$ | $0.50 \pm 0.0132$ | $0.66 \pm 0.0952$ | $0.86 \pm 0.0172$ | $0.59 \pm 0.2818$ | $0.95 \pm 0.0613$ |
| give way + stop | $0.98 \pm 0.0056$ | $0.81 \pm 0.0415$ | $0.70 \pm 0.1508$ | $0.83 \pm 0.0929$ | $1.00 \pm 0.1991$ | $0.70 \pm 0.0922$ |
| danger + construction warning | $0.91 \pm 0.0099$ | $0.68 \pm 0.0121$ | $0.32 \pm 0.0889$ | $0.90 \pm 0.0137$ | $2.82 \pm 0.2851$ | $0.97 \pm 0.0210$ |
| all restriction ends signs | $1.00 \pm 0.0000$ | $0.93 \pm 0.0127$ | $0.60 \pm 0.0791$ | $1.00 \pm 0.0039$ | $0.21 \pm 0.0519$ | $0.94 \pm 0.0221$ |
| all speed limit signs | $0.96 \pm 0.0174$ | $0.79 \pm 0.0075$ | $0.51 \pm 0.0419$ | $0.95 \pm 0.0175$ | $0.29 \pm 0.0730$ | $0.97 \pm 0.0469$ |
| all blue signs | $1.00 \pm 0.0011$ | $0.94 \pm 0.0165$ | $0.34 \pm 0.0640$ | $0.91 \pm 0.0224$ | $0.38 \pm 0.0667$ | $1.00 \pm 0.0023$ |
| all warning signs | $0.97 \pm 0.0042$ | $0.86 \pm 0.0182$ | $0.33 \pm 0.0692$ | $0.96 \pm 0.0061$ | $1.31 \pm 0.2118$ | $1.00 \pm 0.0036$ |
| mean | $0.93 \pm 0.0563$ | $0.73 \pm 0.1517$ | $0.49 \pm 0.1439$ | $0.90 \pm 0.0508$ | $0.95 \pm 0.7345$ | $0.94 \pm 0.0840$ |

Table 16: AD and explanation performance averaged over 4 random seeds on GTSDB for DSVDD. Each row shows results for a different normal definition.

| Normal | AD | | Explanation | | | |
| | AuROC | Score distance | CF AuROC | Sub. AuROC | $FID_N$ | Concept Acc |
| --- | --- | --- | --- | --- | --- | --- |
| speed limit 30 + 50 | $0.53 \pm 0.0718$ | $0.06 \pm 0.0214$ | $0.56 \pm 0.0583$ | $0.57 \pm 0.0240$ | $1.07 \pm 0.4804$ | $0.95 \pm 0.0439$ |
| speed limit 50 + 70 | $0.55 \pm 0.0487$ | $0.07 \pm 0.0640$ | $0.60 \pm 0.1042$ | $0.57 \pm 0.0485$ | $3.59 \pm 3.8551$ | $0.87 \pm 0.1167$ |
| speed limit 70 + 100 | $0.56 \pm 0.0433$ | $0.02 \pm 0.0108$ | $0.53 \pm 0.1288$ | $0.63 \pm 0.0291$ | $0.34 \pm 0.0187$ | $0.92 \pm 0.0376$ |
| speed limit 100 + 120 | $0.61 \pm 0.0497$ | $0.04 \pm 0.0171$ | $0.53 \pm 0.0625$ | $0.64 \pm 0.0488$ | $0.28 \pm 0.0315$ | $0.95 \pm 0.0302$ |
| give way + stop | $0.49 \pm 0.0673$ | $0.00 \pm 0.0150$ | $0.46 \pm 0.0981$ | $0.49 \pm 0.0725$ | $1.88 \pm 0.5662$ | $0.98 \pm 0.0138$ |
| danger + construction warning | $0.61 \pm 0.0429$ | $0.02 \pm 0.0049$ | $0.59 \pm 0.0402$ | $0.47 \pm 0.0348$ | $3.04 \pm 0.3589$ | $0.90 \pm 0.1063$ |
| all restriction ends signs | $0.70 \pm 0.0860$ | $0.06 \pm 0.0450$ | $0.53 \pm 0.1242$ | $0.69 \pm 0.0862$ | $0.26 \pm 0.1251$ | $0.94 \pm 0.0273$ |
| all speed limit signs | $0.69 \pm 0.0473$ | $0.05 \pm 0.0095$ | $0.57 \pm 0.0533$ | $0.64 \pm 0.0145$ | $0.51 \pm 0.1984$ | $0.98 \pm 0.0182$ |
| all blue signs | $0.51 \pm 0.1008$ | $0.02 \pm 0.0161$ | $0.49 \pm 0.0985$ | $0.64 \pm 0.0117$ | $0.20 \pm 0.0484$ | $0.86 \pm 0.0565$ |
| all warning signs | $0.56 \pm 0.0242$ | $0.01 \pm 0.0087$ | $0.46 \pm 0.0616$ | $0.51 \pm 0.0484$ | $1.93 \pm 0.5590$ | $1.00 \pm 0.0034$ |
| mean | $0.58 \pm 0.0668$ | $0.04 \pm 0.0233$ | $0.53 \pm 0.0478$ | $0.58 \pm 0.0699$ | $1.31 \pm 1.1807$ | $0.93 \pm 0.0453$ |

Table 17: AD and explanation performance averaged over 4 random seeds on MNIST for BCE (OE). Each row shows results for a different normal definition.

| Normal | AD | | Explanation | | | |
|---|---|---|---|---|---|---|
| | AuROC | Score distance | CF AuROC | Sub. AuROC | FID$_N$ | Concept Acc |
| zero+one | 0.97 ± 0.0062 | 0.51 ± 0.0596 | 0.79 ± 0.0864 | 0.45 ± 0.0944 | 1.00 ± 0.0674 | 0.98 ± 0.0154 |
| zero+two | 0.95 ± 0.0129 | 0.44 ± 0.0694 | 0.82 ± 0.0696 | 0.59 ± 0.0292 | 0.77 ± 0.0372 | 0.95 ± 0.0520 |
| one+two | 0.94 ± 0.0188 | 0.46 ± 0.0688 | 0.74 ± 0.0251 | 0.40 ± 0.0411 | 1.25 ± 0.0237 | 0.99 ± 0.0101 |
| one+three | 0.95 ± 0.0097 | 0.45 ± 0.0222 | 0.70 ± 0.0433 | 0.56 ± 0.0241 | 1.18 ± 0.0250 | 0.97 ± 0.0192 |
| two+three | 0.97 ± 0.0095 | 0.56 ± 0.0667 | 0.76 ± 0.0720 | 0.79 ± 0.0188 | 0.51 ± 0.0498 | 0.99 ± 0.0131 |
| two+four | 0.89 ± 0.0196 | 0.35 ± 0.0551 | 0.75 ± 0.0415 | 0.42 ± 0.0421 | 0.83 ± 0.0824 | 1.00 ± 0.0017 |
| three+four | 0.91 ± 0.0070 | 0.33 ± 0.0250 | 0.81 ± 0.0290 | 0.58 ± 0.0415 | 0.85 ± 0.0359 | 0.93 ± 0.0687 |
| three+five | 0.95 ± 0.0058 | 0.48 ± 0.0487 | 0.74 ± 0.0213 | 0.67 ± 0.0515 | 0.43 ± 0.0501 | 0.95 ± 0.0360 |
| four+five | 0.90 ± 0.0259 | 0.30 ± 0.0148 | 0.83 ± 0.0474 | 0.40 ± 0.0485 | 0.92 ± 0.0715 | 0.82 ± 0.1926 |
| four+six | 0.95 ± 0.0052 | 0.57 ± 0.0364 | 0.77 ± 0.0333 | 0.63 ± 0.0650 | 0.67 ± 0.1253 | 0.98 ± 0.0277 |
| five+six | 0.97 ± 0.0063 | 0.60 ± 0.0319 | 0.82 ± 0.0672 | 0.63 ± 0.0514 | 0.55 ± 0.0666 | 0.91 ± 0.0797 |
| five+seven | 0.88 ± 0.0228 | 0.40 ± 0.0453 | 0.76 ± 0.0546 | 0.59 ± 0.0416 | 1.02 ± 0.0697 | 0.94 ± 0.0361 |
| six+seven | 0.94 ± 0.0143 | 0.44 ± 0.0618 | 0.85 ± 0.0437 | 0.66 ± 0.0622 | 0.92 ± 0.1281 | 0.82 ± 0.1436 |
| six+eight | 0.95 ± 0.0145 | 0.45 ± 0.0398 | 0.81 ± 0.0474 | 0.63 ± 0.0608 | 0.38 ± 0.0205 | 0.96 ± 0.0539 |
| seven+eight | 0.87 ± 0.0208 | 0.33 ± 0.0300 | 0.73 ± 0.0562 | 0.70 ± 0.0264 | 0.90 ± 0.0669 | 0.91 ± 0.0795 |
| seven+nine | 0.95 ± 0.0209 | 0.58 ± 0.0374 | 0.77 ± 0.0628 | 0.88 ± 0.0201 | 0.94 ± 0.1804 | 0.86 ± 0.1010 |
| eight+nine | 0.93 ± 0.0189 | 0.42 ± 0.0492 | 0.80 ± 0.0483 | 0.83 ± 0.0144 | 0.48 ± 0.0423 | 0.93 ± 0.1050 |
| eight+zero | 0.93 ± 0.0100 | 0.39 ± 0.0219 | 0.77 ± 0.0908 | 0.69 ± 0.0240 | 0.46 ± 0.0200 | 0.98 ± 0.0177 |
| nine+zero | 0.95 ± 0.0047 | 0.49 ± 0.0184 | 0.85 ± 0.0398 | 0.77 ± 0.0424 | 0.54 ± 0.0610 | 0.92 ± 0.0678 |
| nine+one | 0.93 ± 0.0157 | 0.39 ± 0.0365 | 0.73 ± 0.0944 | 0.57 ± 0.0461 | 1.09 ± 0.0559 | 0.97 ± 0.0191 |
| mean | 0.93 ± 0.0283 | 0.45 ± 0.0868 | 0.78 ± 0.0412 | 0.62 ± 0.1325 | 0.78 ± 0.2596 | 0.94 ± 0.0512 |

Table 18: AD and explanation performance averaged over 4 random seeds on MNIST for HSC (OE). Each row shows results for a different normal definition.

| Normal | AD | | Explanation | | | |
|---|---|---|---|---|---|---|
| | AuROC | Score distance | CF AuROC | Sub. AuROC | FID$_N$ | Concept Acc |
| zero+one | 0.98 ± 0.0056 | 0.53 ± 0.0871 | 0.88 ± 0.0450 | 0.46 ± 0.0714 | 1.13 ± 0.0433 | 0.92 ± 0.1256 |
| zero+two | 0.95 ± 0.0120 | 0.52 ± 0.0508 | 0.87 ± 0.0267 | 0.39 ± 0.0644 | 0.96 ± 0.0884 | 0.94 ± 0.0697 |
| one+two | 0.96 ± 0.0061 | 0.48 ± 0.0493 | 0.83 ± 0.0163 | 0.46 ± 0.1134 | 1.23 ± 0.0469 | 0.95 ± 0.0382 |
| one+three | 0.95 ± 0.0081 | 0.51 ± 0.0142 | 0.84 ± 0.0519 | 0.55 ± 0.0545 | 1.24 ± 0.0717 | 0.85 ± 0.2038 |
| two+three | 0.95 ± 0.0116 | 0.58 ± 0.0371 | 0.74 ± 0.0500 | 0.59 ± 0.0706 | 0.73 ± 0.1404 | 0.87 ± 0.1477 |
| two+four | 0.86 ± 0.0132 | 0.33 ± 0.0276 | 0.77 ± 0.0338 | 0.39 ± 0.0131 | 0.92 ± 0.0227 | 0.98 ± 0.0168 |
| three+four | 0.87 ± 0.0190 | 0.34 ± 0.0472 | 0.73 ± 0.0515 | 0.55 ± 0.0355 | 0.87 ± 0.0564 | 0.87 ± 0.1123 |
| three+five | 0.93 ± 0.0294 | 0.50 ± 0.0450 | 0.80 ± 0.0902 | 0.54 ± 0.0523 | 0.54 ± 0.0908 | 0.85 ± 0.1274 |
| four+five | 0.87 ± 0.0160 | 0.33 ± 0.0228 | 0.86 ± 0.0449 | 0.42 ± 0.0571 | 1.35 ± 0.4027 | 0.58 ± 0.0420 |
| four+six | 0.95 ± 0.0128 | 0.55 ± 0.0598 | 0.82 ± 0.0360 | 0.50 ± 0.1191 | 0.82 ± 0.0307 | 0.97 ± 0.0223 |
| five+six | 0.95 ± 0.0058 | 0.57 ± 0.0471 | 0.83 ± 0.0505 | 0.54 ± 0.0711 | 1.03 ± 0.3435 | 0.83 ± 0.0677 |
| five+seven | 0.89 ± 0.0022 | 0.40 ± 0.0223 | 0.83 ± 0.0281 | 0.58 ± 0.0241 | 1.33 ± 0.2102 | 0.80 ± 0.1326 |
| six+seven | 0.92 ± 0.0166 | 0.43 ± 0.0602 | 0.81 ± 0.0535 | 0.54 ± 0.0695 | 1.02 ± 0.3005 | 0.87 ± 0.0852 |
| six+eight | 0.94 ± 0.0031 | 0.44 ± 0.0373 | 0.81 ± 0.0184 | 0.51 ± 0.0417 | 0.51 ± 0.1461 | 0.88 ± 0.0918 |
| seven+eight | 0.90 ± 0.0090 | 0.42 ± 0.0328 | 0.78 ± 0.0331 | 0.66 ± 0.0287 | 1.14 ± 0.0710 | 0.91 ± 0.0864 |
| seven+nine | 0.96 ± 0.0034 | 0.63 ± 0.0163 | 0.85 ± 0.0637 | 0.81 ± 0.0430 | 1.17 ± 0.2448 | 0.65 ± 0.2011 |
| eight+nine | 0.93 ± 0.0049 | 0.44 ± 0.0268 | 0.83 ± 0.0483 | 0.69 ± 0.0317 | 0.67 ± 0.1301 | 0.87 ± 0.1908 |
| eight+zero | 0.93 ± 0.0075 | 0.44 ± 0.0215 | 0.83 ± 0.0602 | 0.55 ± 0.0547 | 0.80 ± 0.4024 | 0.85 ± 0.1161 |
| nine+zero | 0.94 ± 0.0052 | 0.48 ± 0.0601 | 0.85 ± 0.0379 | 0.61 ± 0.0466 | 0.65 ± 0.0405 | 0.77 ± 0.1480 |
| nine+one | 0.95 ± 0.0119 | 0.44 ± 0.0212 | 0.83 ± 0.0464 | 0.60 ± 0.0340 | 1.13 ± 0.0206 | 0.92 ± 0.0678 |
| mean | 0.93 ± 0.0332 | 0.47 ± 0.0809 | 0.82 ± 0.0378 | 0.55 ± 0.0987 | 0.96 ± 0.2502 | 0.86 ± 0.0963 |

Table 19: AD and explanation performance averaged over 4 random seeds on MNIST for DSVDD. Each row shows results for a different normal definition.

| | AD | | Explanation | | | |
|---|---|---|---|---|---|---|
| Normal | AuROC | Score distance | CF AuROC | Sub. AuROC | FID$_N$ | Concept Acc |
| zero+one | 0.93 ± 0.0323 | 0.00 ± 0.0018 | 0.90 ± 0.0393 | 0.57 ± 0.0150 | 1.05 ± 0.1323 | 0.97 ± 0.0254 |
| zero+two | 0.71 ± 0.1290 | 0.00 ± 0.0015 | 0.70 ± 0.1319 | 0.36 ± 0.0439 | 0.99 ± 0.0301 | 0.54 ± 0.2298 |
| one+two | 0.73 ± 0.0542 | 0.00 ± 0.0003 | 0.73 ± 0.0648 | 0.38 ± 0.0584 | 1.16 ± 0.0277 | 0.92 ± 0.0666 |
| one+three | 0.77 ± 0.0422 | 0.00 ± 0.0002 | 0.78 ± 0.0470 | 0.43 ± 0.1285 | 1.13 ± 0.0103 | 0.87 ± 0.1073 |
| two+three | 0.69 ± 0.0508 | 0.00 ± 0.0015 | 0.67 ± 0.0495 | 0.38 ± 0.1011 | 0.86 ± 0.0373 | 0.81 ± 0.2033 |
| two+four | 0.85 ± 0.0253 | 0.00 ± 0.0009 | 0.80 ± 0.0380 | 0.39 ± 0.0484 | 0.75 ± 0.1440 | 0.85 ± 0.2204 |
| three+four | 0.77 ± 0.0716 | 0.00 ± 0.0015 | 0.73 ± 0.0736 | 0.46 ± 0.0377 | 0.92 ± 0.0610 | 0.72 ± 0.2467 |
| three+five | 0.66 ± 0.0275 | 0.00 ± 0.0003 | 0.66 ± 0.0346 | 0.43 ± 0.0459 | 0.86 ± 0.0218 | 0.76 ± 0.1619 |
| four+five | 0.71 ± 0.1077 | 0.00 ± 0.0026 | 0.70 ± 0.0907 | 0.41 ± 0.0192 | 0.98 ± 0.0285 | 0.71 ± 0.0798 |
| four+six | 0.81 ± 0.0719 | 0.01 ± 0.0037 | 0.80 ± 0.0915 | 0.37 ± 0.0288 | 1.03 ± 0.0127 | 0.86 ± 0.1675 |
| five+six | 0.72 ± 0.0814 | 0.00 ± 0.0028 | 0.70 ± 0.0749 | 0.41 ± 0.0568 | 0.93 ± 0.0151 | 0.73 ± 0.1704 |
| five+seven | 0.72 ± 0.0564 | 0.00 ± 0.0009 | 0.69 ± 0.0281 | 0.44 ± 0.0658 | 0.96 ± 0.0983 | 0.85 ± 0.1442 |
| six+seven | 0.84 ± 0.0609 | 0.00 ± 0.0015 | 0.79 ± 0.0271 | 0.41 ± 0.0469 | 1.13 ± 0.0494 | 0.94 ± 0.0260 |
| six+eight | 0.78 ± 0.0681 | 0.00 ± 0.0013 | 0.75 ± 0.0787 | 0.44 ± 0.0241 | 0.93 ± 0.1650 | 0.79 ± 0.1834 |
| seven+eight | 0.70 ± 0.0095 | 0.00 ± 0.0002 | 0.70 ± 0.0046 | 0.39 ± 0.0721 | 1.12 ± 0.0105 | 0.95 ± 0.0364 |
| seven+nine | 0.74 ± 0.0744 | 0.00 ± 0.0020 | 0.75 ± 0.0758 | 0.38 ± 0.0345 | 1.10 ± 0.0419 | 0.72 ± 0.1768 |
| eight+nine | 0.69 ± 0.0688 | 0.00 ± 0.0006 | 0.68 ± 0.0712 | 0.42 ± 0.0329 | 0.95 ± 0.1594 | 0.97 ± 0.0480 |
| eight+zero | 0.66 ± 0.0560 | 0.00 ± 0.0009 | 0.65 ± 0.0630 | 0.37 ± 0.0299 | 1.05 ± 0.0253 | 0.82 ± 0.1814 |
| nine+zero | 0.72 ± 0.0834 | 0.00 ± 0.0016 | 0.67 ± 0.1228 | 0.46 ± 0.0408 | 0.99 ± 0.1008 | 0.65 ± 0.3174 |
| nine+one | 0.84 ± 0.0555 | 0.00 ± 0.0010 | 0.85 ± 0.0489 | 0.42 ± 0.1575 | 1.13 ± 0.0173 | 0.91 ± 0.0509 |
| mean | 0.75 ± 0.0712 | 0.00 ± 0.0013 | 0.73 ± 0.0649 | 0.42 ± 0.0450 | 1.00 ± 0.1074 | 0.82 ± 0.1132 |

Table 20: AD and explanation performance averaged over 4 random seeds on CIFAR-10 for BCE OE. Each row shows results for a different normal definition.

| | AD | | Explanation | | | |
|---|---|---|---|---|---|---|
| Normal | AuROC | Score distance | CF AuROC | Sub. AuROC | FID$_N$ | Concept Acc |
| airplane+automobile | 0.96 ± 0.0024 | 0.79 ± 0.0066 | 0.59 ± 0.0300 | 0.66 ± 0.0187 | 1.04 ± 0.0824 | 0.75 ± 0.1067 |
| airplane+bird | 0.92 ± 0.0017 | 0.68 ± 0.0043 | 0.45 ± 0.0226 | 0.61 ± 0.0087 | 1.34 ± 0.2551 | 0.88 ± 0.1167 |
| automobile+bird | 0.93 ± 0.0023 | 0.70 ± 0.0029 | 0.57 ± 0.0340 | 0.59 ± 0.0264 | 1.79 ± 0.0164 | 0.73 ± 0.2012 |
| automobile+cat | 0.90 ± 0.0038 | 0.61 ± 0.0005 | 0.46 ± 0.0113 | 0.54 ± 0.0060 | 1.73 ± 0.0686 | 0.87 ± 0.0738 |
| bird+cat | 0.87 ± 0.0022 | 0.53 ± 0.0019 | 0.35 ± 0.0207 | 0.54 ± 0.0140 | 1.19 ± 0.1377 | 0.81 ± 0.1128 |
| bird+deer | 0.92 ± 0.0004 | 0.64 ± 0.0046 | 0.39 ± 0.0233 | 0.53 ± 0.0069 | 0.92 ± 0.0889 | 0.97 ± 0.0038 |
| cat+deer | 0.90 ± 0.0025 | 0.58 ± 0.0077 | 0.39 ± 0.0301 | 0.53 ± 0.0148 | 0.94 ± 0.0475 | 0.89 ± 0.1547 |
| cat+dog | 0.91 ± 0.0023 | 0.59 ± 0.0108 | 0.30 ± 0.0103 | 0.58 ± 0.0099 | 0.91 ± 0.0472 | 0.81 ± 0.1551 |
| deer+dog | 0.92 ± 0.0006 | 0.64 ± 0.0040 | 0.42 ± 0.0333 | 0.55 ± 0.0137 | 0.88 ± 0.0511 | 0.93 ± 0.0495 |
| deer+frog | 0.94 ± 0.0014 | 0.70 ± 0.0042 | 0.49 ± 0.0381 | 0.52 ± 0.0124 | 0.76 ± 0.0422 | 0.82 ± 0.1905 |
| dog+frog | 0.93 ± 0.0010 | 0.67 ± 0.0053 | 0.46 ± 0.0181 | 0.56 ± 0.0121 | 0.93 ± 0.0769 | 0.94 ± 0.0597 |
| dog+horse | 0.95 ± 0.0022 | 0.71 ± 0.0056 | 0.50 ± 0.0085 | 0.58 ± 0.0106 | 1.01 ± 0.0391 | 0.89 ± 0.1399 |
| frog+horse | 0.96 ± 0.0007 | 0.76 ± 0.0080 | 0.55 ± 0.0314 | 0.56 ± 0.0170 | 1.03 ± 0.0501 | 0.81 ± 0.1722 |
| frog+ship | 0.95 ± 0.0010 | 0.76 ± 0.0046 | 0.53 ± 0.0225 | 0.62 ± 0.0188 | 1.06 ± 0.2823 | 0.88 ± 0.0802 |
| horse+ship | 0.97 ± 0.0010 | 0.80 ± 0.0047 | 0.58 ± 0.0259 | 0.61 ± 0.0420 | 0.95 ± 0.1126 | 0.97 ± 0.0323 |
| horse+truck | 0.96 ± 0.0008 | 0.77 ± 0.0046 | 0.56 ± 0.0293 | 0.60 ± 0.0195 | 1.08 ± 0.0864 | 0.87 ± 0.1812 |
| ship+truck | 0.96 ± 0.0011 | 0.77 ± 0.0059 | 0.54 ± 0.0200 | 0.62 ± 0.0171 | 0.78 ± 0.0594 | 0.93 ± 0.1109 |
| ship+airplane | 0.97 ± 0.0008 | 0.80 ± 0.0044 | 0.52 ± 0.0392 | 0.71 ± 0.0113 | 0.77 ± 0.1048 | 0.97 ± 0.0441 |
| truck+airplane | 0.95 ± 0.0008 | 0.75 ± 0.0027 | 0.55 ± 0.0137 | 0.61 ± 0.0370 | 0.93 ± 0.0557 | 0.73 ± 0.1478 |
| truck+automobile | 0.98 ± 0.0010 | 0.85 ± 0.0041 | 0.62 ± 0.0429 | 0.60 ± 0.0240 | 0.75 ± 0.0793 | 0.80 ± 0.1978 |
| mean | 0.94 ± 0.0266 | 0.71 ± 0.0839 | 0.49 ± 0.0847 | 0.59 ± 0.0460 | 1.04 ± 0.2794 | 0.86 ± 0.0745 |

Table 21: AD and explanation performance averaged over 4 random seeds on CIFAR-10 for HSC OE. Each row shows results for a different normal definition.

| Normal | AD | | Explanation | | | |
| | AuROC | Score distance | CF AuROC | Sub. AuROC | $FID_N$ | Concept Acc |
|---|---|---|---|---|---|---|
| airplane+automobile | 0.96 ± 0.0005 | 0.75 ± 0.0017 | 0.51 ± 0.0900 | 0.54 ± 0.0163 | 2.14 ± 0.0882 | 0.99 ± 0.0164 |
| airplane+bird | 0.93 ± 0.0012 | 0.67 ± 0.0024 | 0.44 ± 0.0439 | 0.52 ± 0.0059 | 2.21 ± 0.1630 | 1.00 ± 0.0002 |
| automobile+bird | 0.92 ± 0.0029 | 0.66 ± 0.0065 | 0.45 ± 0.0424 | 0.51 ± 0.0065 | 4.12 ± 1.1471 | 1.00 ± 0.0001 |
| automobile+cat | 0.91 ± 0.0011 | 0.62 ± 0.0054 | 0.53 ± 0.0285 | 0.50 ± 0.0023 | 3.10 ± 0.3450 | 1.00 ± 0.0011 |
| bird+cat | 0.87 ± 0.0019 | 0.47 ± 0.0046 | 0.32 ± 0.0328 | 0.53 ± 0.0401 | 3.34 ± 1.0615 | 1.00 ± 0.0002 |
| bird+deer | 0.92 ± 0.0026 | 0.63 ± 0.0097 | 0.38 ± 0.0144 | 0.54 ± 0.0248 | 3.49 ± 0.1061 | 1.00 ± 0.0012 |
| cat+deer | 0.90 ± 0.0017 | 0.54 ± 0.0053 | 0.35 ± 0.0228 | 0.52 ± 0.0166 | 2.58 ± 0.1145 | 1.00 ± 0.0000 |
| cat+dog | 0.93 ± 0.0018 | 0.59 ± 0.0085 | 0.39 ± 0.0252 | 0.52 ± 0.0042 | 1.97 ± 0.0935 | 1.00 ± 0.0003 |
| deer+dog | 0.92 ± 0.0017 | 0.60 ± 0.0095 | 0.38 ± 0.0401 | 0.52 ± 0.0107 | 2.44 ± 0.5742 | 0.96 ± 0.0734 |
| deer+frog | 0.95 ± 0.0011 | 0.68 ± 0.0010 | 0.42 ± 0.0065 | 0.56 ± 0.0535 | 2.27 ± 0.0879 | 1.00 ± 0.0002 |
| dog+frog | 0.93 ± 0.0014 | 0.63 ± 0.0045 | 0.43 ± 0.0110 | 0.51 ± 0.0036 | 2.53 ± 0.1879 | 1.00 ± 0.0001 |
| dog+horse | 0.96 ± 0.0003 | 0.70 ± 0.0064 | 0.44 ± 0.0062 | 0.52 ± 0.0190 | 3.22 ± 0.1861 | 1.00 ± 0.0001 |
| frog+horse | 0.96 ± 0.0015 | 0.73 ± 0.0027 | 0.48 ± 0.0143 | 0.52 ± 0.0176 | 2.75 ± 0.3541 | 1.00 ± 0.0001 |
| frog+ship | 0.96 ± 0.0009 | 0.75 ± 0.0084 | 0.48 ± 0.0313 | 0.56 ± 0.0346 | 3.29 ± 0.6680 | 1.00 ± 0.0001 |
| horse+ship | 0.96 ± 0.0007 | 0.77 ± 0.0036 | 0.40 ± 0.0675 | 0.53 ± 0.0124 | 1.87 ± 0.0485 | 1.00 ± 0.0005 |
| horse+truck | 0.95 ± 0.0016 | 0.73 ± 0.0074 | 0.50 ± 0.0339 | 0.53 ± 0.0520 | 2.93 ± 0.8821 | 1.00 ± 0.0011 |
| ship+truck | 0.96 ± 0.0005 | 0.76 ± 0.0051 | 0.41 ± 0.0426 | 0.57 ± 0.0625 | 1.73 ± 0.0526 | 0.99 ± 0.0075 |
| ship+airplane | 0.97 ± 0.0013 | 0.80 ± 0.0037 | 0.53 ± 0.0811 | 0.55 ± 0.0359 | 1.65 ± 0.2366 | 0.98 ± 0.0247 |
| truck+airplane | 0.95 ± 0.0020 | 0.72 ± 0.0041 | 0.46 ± 0.0542 | 0.53 ± 0.0176 | 1.85 ± 0.1448 | 0.97 ± 0.0579 |
| truck+automobile | 0.99 ± 0.0004 | 0.85 ± 0.0067 | 0.60 ± 0.0790 | 0.53 ± 0.0340 | 1.49 ± 0.1063 | 0.90 ± 0.1301 |
| mean | 0.94 ± 0.0270 | 0.68 ± 0.0883 | 0.44 ± 0.0666 | 0.53 ± 0.0175 | 2.55 ± 0.6970 | 0.99 ± 0.0244 |

Table 22: AD and explanation performance averaged over 4 random seeds on CIFAR-10 for DSVDD. Each row shows results for a different normal definition.

| Normal | AD | | Explanation | | | |
| | AuROC | Score distance | CF AuROC | Sub. AuROC | $FID_N$ | Concept Acc |
|---|---|---|---|---|---|---|
| airplane+automobile | 0.50 ± 0.0357 | 0.00 ± 0.0002 | 0.48 ± 0.0517 | 0.46 ± 0.0260 | 1.20 ± 0.0111 | 0.84 ± 0.1424 |
| airplane+bird | 0.49 ± 0.0111 | 0.00 ± 0.0005 | 0.46 ± 0.0219 | 0.49 ± 0.0448 | 1.27 ± 0.0950 | 0.93 ± 0.0503 |
| automobile+bird | 0.49 ± 0.0145 | 0.00 ± 0.0002 | 0.49 ± 0.0081 | 0.49 ± 0.0184 | 1.23 ± 0.0524 | 0.93 ± 0.0859 |
| automobile+cat | 0.50 ± 0.0148 | 0.00 ± 0.0007 | 0.48 ± 0.0153 | 0.47 ± 0.0251 | 1.22 ± 0.0567 | 0.90 ± 0.0745 |
| bird+cat | 0.53 ± 0.0162 | 0.00 ± 0.0003 | 0.51 ± 0.0344 | 0.50 ± 0.0033 | 1.08 ± 0.0223 | 0.98 ± 0.0223 |
| bird+deer | 0.56 ± 0.0278 | 0.00 ± 0.0003 | 0.54 ± 0.0345 | 0.51 ± 0.0122 | 0.97 ± 0.0304 | 0.97 ± 0.0183 |
| cat+deer | 0.56 ± 0.0418 | 0.00 ± 0.0008 | 0.54 ± 0.0486 | 0.53 ± 0.0228 | 1.02 ± 0.0201 | 0.95 ± 0.0201 |
| cat+dog | 0.52 ± 0.0105 | 0.00 ± 0.0011 | 0.49 ± 0.0332 | 0.49 ± 0.0148 | 1.06 ± 0.0168 | 0.91 ± 0.0690 |
| deer+dog | 0.55 ± 0.0213 | 0.00 ± 0.0030 | 0.51 ± 0.0377 | 0.53 ± 0.0211 | 1.10 ± 0.0348 | 0.89 ± 0.1620 |
| deer+frog | 0.57 ± 0.1151 | 0.01 ± 0.0046 | 0.53 ± 0.1167 | 0.59 ± 0.0516 | 0.87 ± 0.0342 | 0.93 ± 0.0919 |
| dog+frog | 0.60 ± 0.0431 | 0.00 ± 0.0034 | 0.60 ± 0.0514 | 0.53 ± 0.0323 | 0.95 ± 0.0188 | 0.87 ± 0.0848 |
| dog+horse | 0.53 ± 0.0102 | 0.00 ± 0.0006 | 0.49 ± 0.0408 | 0.49 ± 0.0178 | 1.17 ± 0.0254 | 0.92 ± 0.0427 |
| frog+horse | 0.60 ± 0.0398 | 0.01 ± 0.0048 | 0.56 ± 0.0160 | 0.57 ± 0.0228 | 1.07 ± 0.0079 | 0.99 ± 0.0030 |
| frog+ship | 0.52 ± 0.0144 | 0.00 ± 0.0004 | 0.50 ± 0.0326 | 0.53 ± 0.0188 | 1.08 ± 0.0331 | 0.97 ± 0.0261 |
| horse+ship | 0.49 ± 0.0374 | 0.00 ± 0.0002 | 0.48 ± 0.0409 | 0.48 ± 0.0077 | 1.17 ± 0.0563 | 0.96 ± 0.0209 |
| horse+truck | 0.50 ± 0.0346 | 0.00 ± 0.0006 | 0.51 ± 0.0287 | 0.46 ± 0.0147 | 1.21 ± 0.0579 | 0.88 ± 0.1041 |
| ship+truck | 0.47 ± 0.0265 | 0.00 ± 0.0003 | 0.49 ± 0.0195 | 0.46 ± 0.0201 | 1.05 ± 0.0330 | 0.96 ± 0.0365 |
| ship+airplane | 0.50 ± 0.0246 | 0.00 ± 0.0002 | 0.48 ± 0.0400 | 0.42 ± 0.0326 | 1.10 ± 0.0722 | 0.87 ± 0.1070 |
| truck+airplane | 0.48 ± 0.0545 | 0.00 ± 0.0004 | 0.48 ± 0.0460 | 0.46 ± 0.0205 | 1.15 ± 0.0309 | 0.94 ± 0.0497 |
| truck+automobile | 0.51 ± 0.0279 | 0.00 ± 0.0009 | 0.52 ± 0.0356 | 0.45 ± 0.0143 | 1.06 ± 0.0331 | 0.86 ± 0.1105 |
| mean | 0.53 ± 0.0356 | 0.00 ± 0.0023 | 0.51 ± 0.0332 | 0.50 ± 0.0414 | 1.10 ± 0.0998 | 0.92 ± 0.0424 |

Table 23: AD and explanation performance averaged over 2 random seeds on ImageNet-Neighbors for BCE (OE). Each row shows results for a different normal definition.

| Normal | AD | | Explanation | | |
| | AuROC | CF AuROC | Sub. AuROC | $FID_N$ | Concept Acc |
| --- | --- | --- | --- | --- | --- |
| airliner | $96.63 \pm 0.22$ | $76.32 \pm 0.82$ | $65.01 \pm 4.57$ | $95.75 \pm 9.65$ | $99.70 \pm 0.20$ |
| ambulance | $98.23 \pm 0.03$ | $83.91 \pm 2.48$ | $63.52 \pm 4.41$ | $105.45 \pm 4.33$ | $99.85 \pm 0.15$ |
| black widow | $90.31 \pm 0.41$ | $68.64 \pm 4.25$ | $56.22 \pm 5.19$ | $100.86 \pm 20.66$ | $86.20 \pm 11.40$ |
| lion | $84.00 \pm 0.07$ | $34.38 \pm 1.10$ | $61.97 \pm 0.11$ | $94.49 \pm 7.87$ | $100.00 \pm 0.00$ |
| zebra | $98.97 \pm 0.02$ | $82.16 \pm 0.65$ | $49.16 \pm 8.66$ | $28.29 \pm 0.43$ | $99.00 \pm 0.70$ |
| mean | $93.63 \pm 5.70$ | $69.08 \pm 18.15$ | $59.18 \pm 5.83$ | $84.97 \pm 28.61$ | $96.95 \pm 5.39$ |

Table 24: AD and explanation performance averaged over 2 random seeds on ImageNet-Neighbors for HSC (OE). Each row shows results for a different normal definition.

| Normal | AD | | Explanation | | |
| | AuROC | CF AuROC | Sub. AuROC | $FID_N$ | Concept Acc |
| --- | --- | --- | --- | --- | --- |
| airliner | $96.70 \pm 0.04$ | $83.04 \pm 0.32$ | $37.43 \pm 0.32$ | $80.26 \pm 2.12$ | $97.30 \pm 2.10$ |
| ambulance | $97.82 \pm 0.01$ | $83.42 \pm 0.67$ | $51.84 \pm 17.77$ | $104.30 \pm 2.86$ | $99.95 \pm 0.05$ |
| black widow | $88.20 \pm 0.20$ | $59.68 \pm 0.52$ | $55.09 \pm 1.12$ | $120.69 \pm 10.51$ | $99.60 \pm 0.40$ |
| lion | $81.35 \pm 0.74$ | $49.83 \pm 7.35$ | $49.20 \pm 5.02$ | $70.58 \pm 11.86$ | $97.85 \pm 1.85$ |
| zebra | $98.78 \pm 0.02$ | $63.84 \pm 3.86$ | $71.63 \pm 1.02$ | $51.17 \pm 6.16$ | $99.70 \pm 0.31$ |
| mean | $92.57 \pm 6.76$ | $67.96 \pm 13.27$ | $53.04 \pm 11.04$ | $85.40 \pm 24.58$ | $98.88 \pm 1.09$ |

# H RANDOM COLLECTION OF GENERATED COUNTERFACTUAL EXAMPLES

In the main paper, we proposed a method to generate counterfactual explanations (CEs) for deep AD. We demonstrated their effectiveness by showing a small fraction of the generated CEs in Section 4.2. Here, we show a larger collection of CEs for all normal definitions. For each normal definition, we randomly selected two samples to serve as examples. Figures 10, 11, and 12 show CEs for Colored-MNIST (C-MNIST) and an AD trained with BCE, HSC, and DSVDD, respectively.

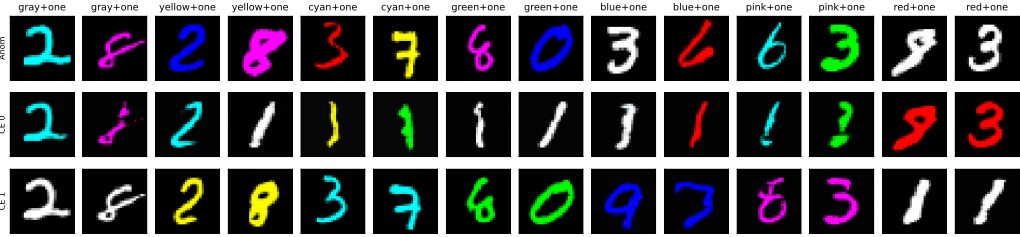

Figure 10: CEs for Col-MNIST and an anomaly detector trained with BCE (OE). For each normal definition, a different detector and CE generator was trained. In each subfigure, the first row shows anomalies, the other two corresponding counterfactuals for two different concepts. Each column is labeled with the corresponding combined normal class at the top.

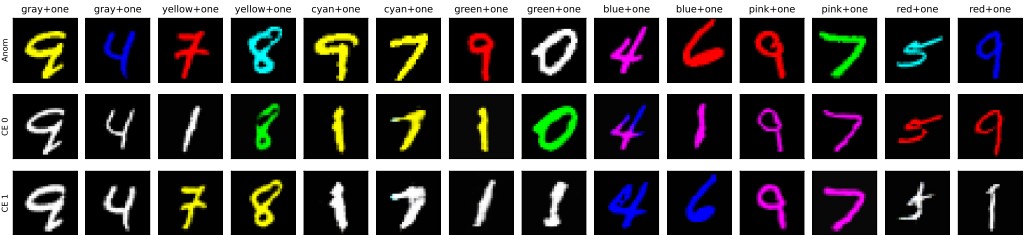

Figure 11: CEs for Col-MNIST and an anomaly detector trained with HSC (OE). For each normal definition, a different detector and CE generator was trained. In each subfigure, the first row shows anomalies, the other two corresponding counterfactuals for two different concepts. Each column is labeled with the corresponding combined normal class at the top.

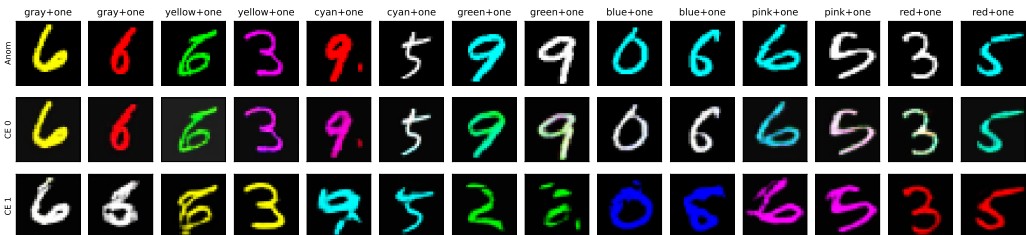

Figure 12: CEs for Col-MNIST and an anomaly detector trained with DSVDD. For each normal definition, a different detector and counterfactual generator was trained. In each subfigure, the first row shows anomalies, the other two corresponding counterfactuals for two different concepts. Each column is labeled with the corresponding combined normal class at the top.

Figures 13, 14, and 15 show CEs for MNIST, single classes being normal, and an AD trained with BCE, HSC, and DSVDD, respectively.

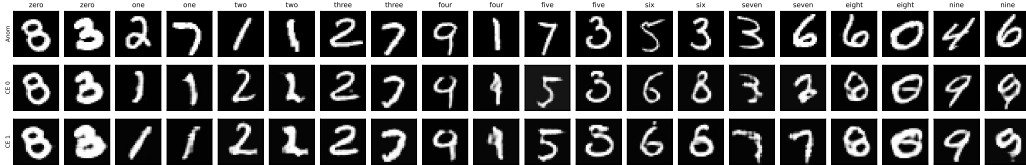

Figure 13: CEs for MNIST, diverse single normal classes, and an anomaly detector trained with BCE (OE). For each normal definition, a different detector and counterfactual generator was trained. In each subfigure, the first row shows anomalies, the other two corresponding counterfactuals for two different concepts. Each column is labeled with the corresponding single normal class at the top.

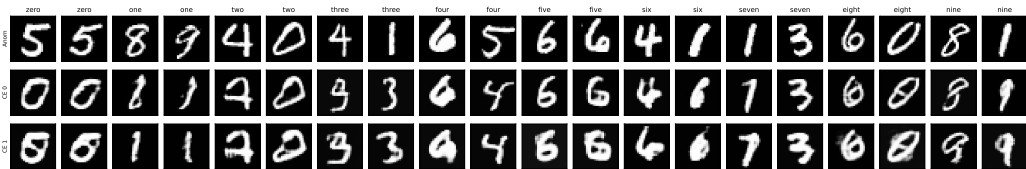

Figure 14: CEs for MNIST, diverse single normal classes, and an anomaly detector trained with HSC (OE). For each normal definition, a different detector and counterfactual generator was trained. In each subfigure, the first row shows anomalies, the other two corresponding counterfactuals for two different concepts. Each column is labeled with the corresponding single normal class at the top.

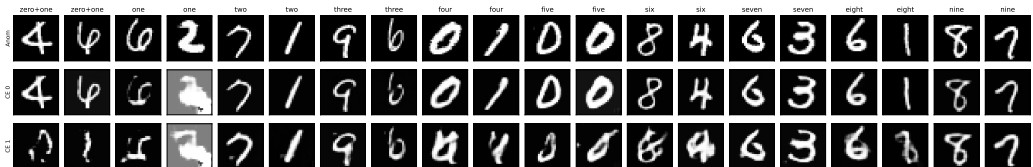

Figure 15: CEs for MNIST, diverse single normal classes, and an anomaly detector trained with DSVDD. For each normal definition, a different detector and counterfactual generator was trained. In each subfigure, the first row shows anomalies, the other two corresponding counterfactuals for two different concepts. Each column is labeled with the corresponding single normal class at the top.

Figures 16, 17, and 18 show CEs for CIFAR-10, single classes being normal, and an AD trained with BCE, HSC, and DSVDD, respectively.

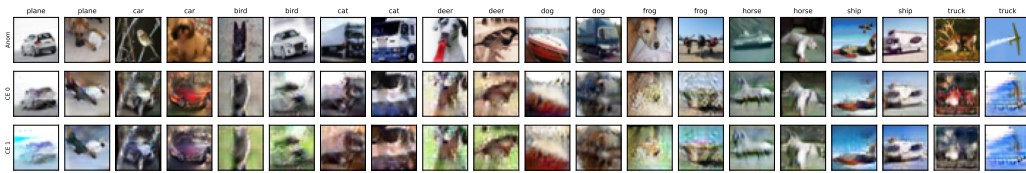

Figure 16: CEs for CIFAR-10, diverse single normal classes, and an anomaly detector trained with BCE (OE). For each normal definition, a different detector and counterfactual generator was trained. In each subfigure, the first row shows anomalies, the other two corresponding counterfactuals for two different concepts. Each column is labeled with the corresponding single normal class at the top.

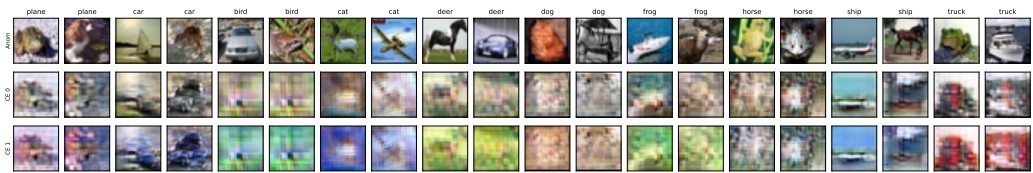

Figure 17: CEs for CIFAR-10, diverse single normal classes, and an anomaly detector trained with HSC (OE). For each normal definition, a different detector and counterfactual generator was trained. In each subfigure, the first row shows anomalies, the other two corresponding counterfactuals for two different concepts. Each column is labeled with the corresponding single normal class at the top.

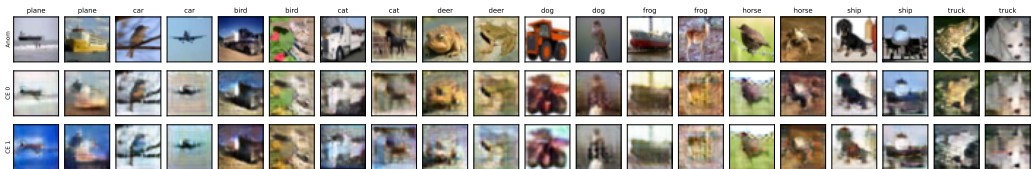

Figure 18: CEs for CIFAR-10, diverse single normal classes, and an anomaly detector trained with DSVDD. For each normal definition, a different detector and counterfactual generator was trained. In each subfigure, the first row shows anomalies, the other two corresponding counterfactuals for two different concepts. Each column is labeled with the corresponding single normal class at the top.

Figures 19, 20, and 21 show CEs for MNIST, class combinations being normal, and an AD trained with BCE, HSC, and DSVDD, respectively.

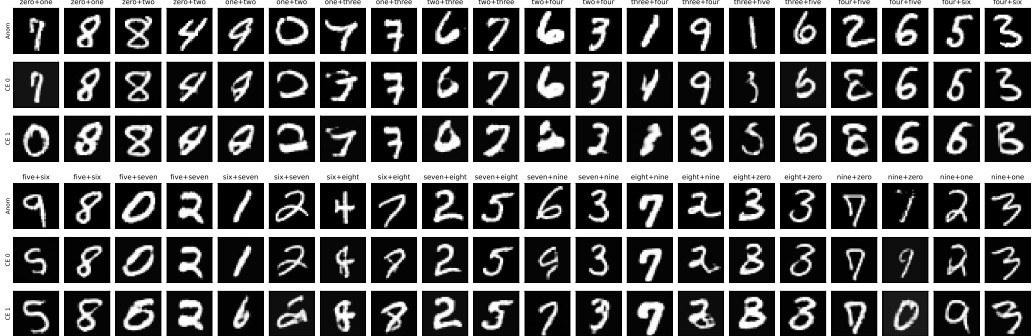

Figure 19: CEs for MNIST, diverse combined normal classes, and an anomaly detector trained with BCE (OE). For each normal definition, a different detector and counterfactual generator was trained. In each subfigure, the first row shows anomalies, the other two corresponding counterfactuals for two different concepts. Each column is labeled with the corresponding combined normal class at the top.

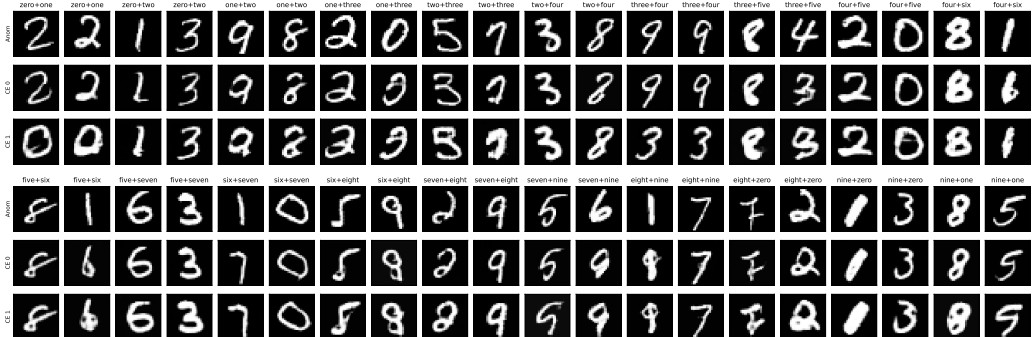

Figure 20: CEs for MNIST, diverse combined normal classes, and an anomaly detector trained with HSC (OE). For each normal definition, a different detector and counterfactual generator was trained. In each subfigure, the first row shows anomalies, the other two corresponding counterfactuals for two different concepts. Each column is labeled with the corresponding combined normal class at the top.

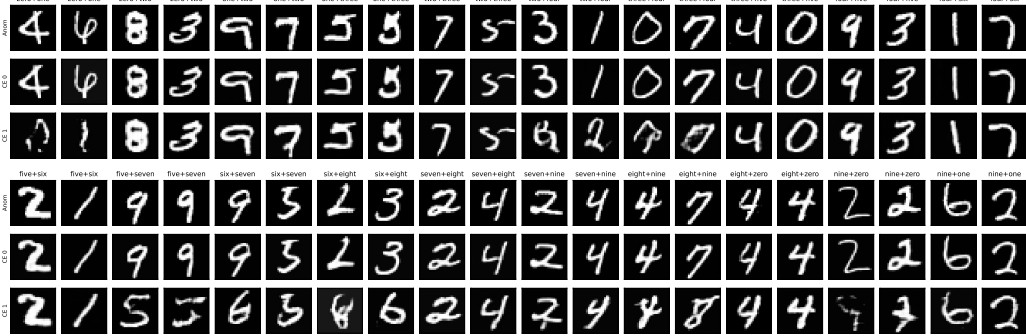

Figure 21: CEs for MNIST, diverse combined normal classes, and an anomaly detector trained with DSVDD. For each normal definition, a different detector and counterfactual generator was trained. In each subfigure, the first row shows anomalies, the other two corresponding counterfactuals for two different concepts. Each column is labeled with the corresponding combined normal class at the top.

Figures 22, 23, and 24 show CEs for CIFAR-10, class combinations being normal, and an AD trained with BCE, HSC, and DSVDD, respectively.

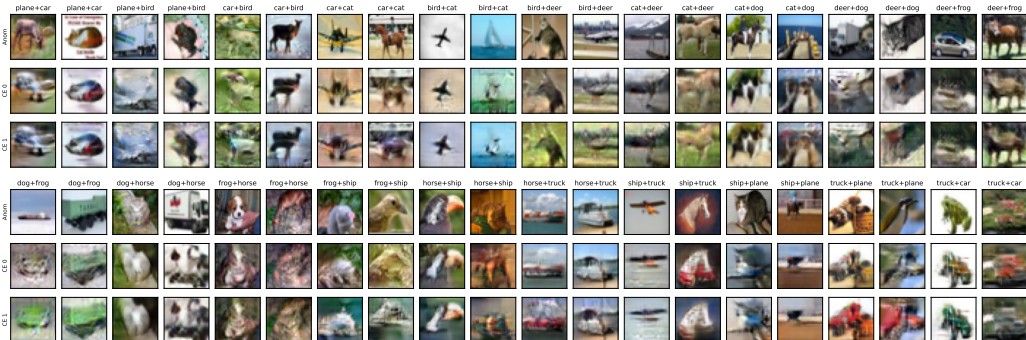

Figure 22: CEs for CIFAR-10, diverse combined normal classes, and an anomaly detector trained with BCE (OE). For each normal definition, a different detector and counterfactual generator was trained. In each subfigure, the first row shows anomalies, the other two corresponding counterfactuals for two different concepts. Each column is labeled with the corresponding combined normal class at the top.

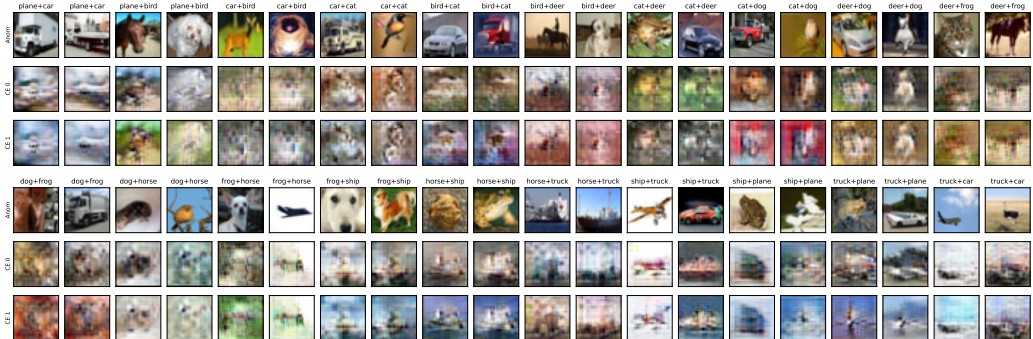

Figure 23: CEs for CIFAR-10, diverse combined normal classes, and an anomaly detector trained with HSC (OE). For each normal definition, a different detector and counterfactual generator was trained. In each subfigure, the first row shows anomalies, the other two corresponding counterfactuals for two different concepts. Each column is labeled with the corresponding combined normal class at the top.

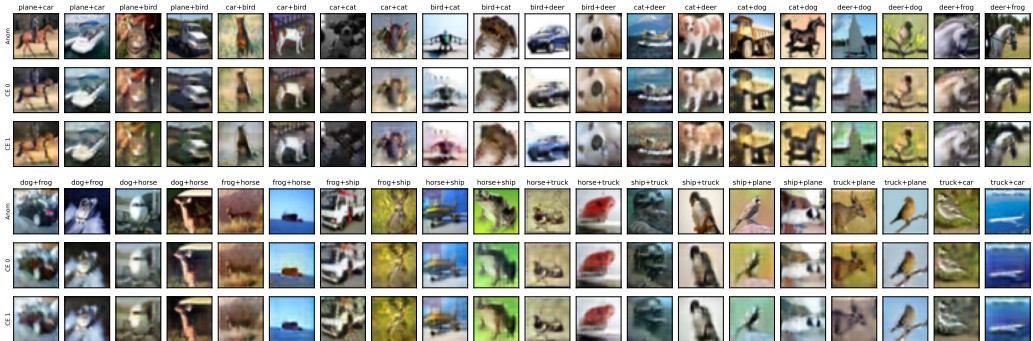

Figure 24: CEs for CIFAR-10, diverse combined normal classes, and an anomaly detector trained with DSVDD. For each normal definition, a different detector and counterfactual generator was trained. In each subfigure, the first row shows anomalies, the other two corresponding counterfactuals for two different concepts. Each column is labeled with the corresponding combined normal class at the top.

Figures 25 and 26 show the CEs for ImageNet-Neighbors, with single classes being normal, and an AD trained with BCE and HSC, respectively.

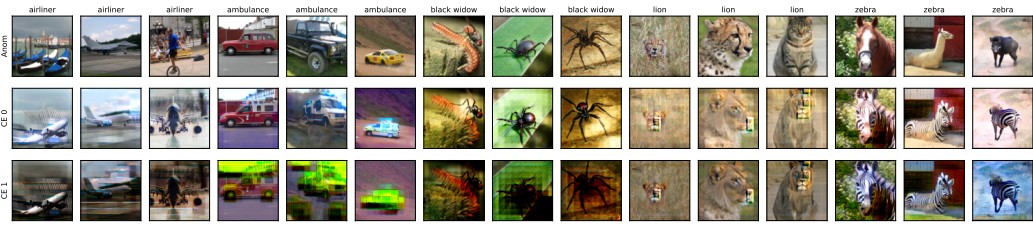

Figure 25: CEs for ImageNet-Neighbors, single normal classes, and an anomaly detector trained with BCE (OE). For each normal definition, a different detector and counterfactual generator was trained. In each subfigure, the first row shows anomalies, the other two corresponding counterfactuals for two different concepts. Each column is labeled with the corresponding normal class at the top.

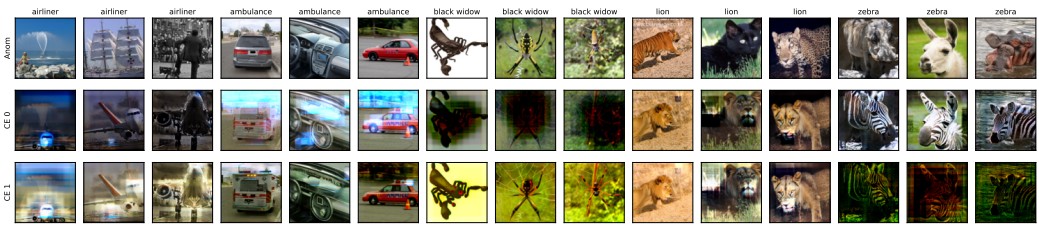

Figure 26: CEs for ImageNet-Neighbors, single normal classes, and an anomaly detector trained with HSC (OE). For each normal definition, a different detector and counterfactual generator was trained. In each subfigure, the first row shows anomalies, the other two corresponding counterfactuals for two different concepts. Each column is labeled with the corresponding normal class at the top.

Figures 27, 28, and 29 show CEs for GTSDB, class combinations being normal, and an AD trained with BCE, HSC, and DSVDD, respectively.

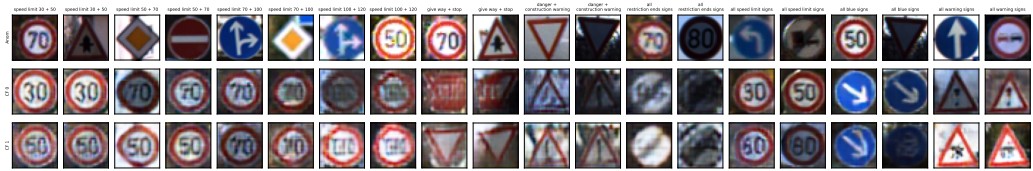

Figure 27: CEs for GTSDB and an anomaly detector trained with BCE OE. For each normal definition, a different detector and counterfactual generator was trained. In each subfigure, the first row shows anomalies, the other two corresponding counterfactuals for two different concepts. Each column is labeled with the corresponding combined normal class at the top.

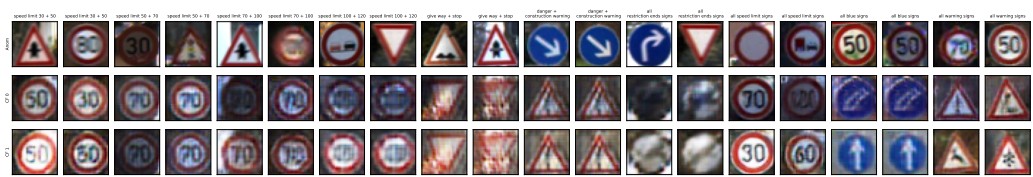

Figure 28: CEs for GTSDB and an anomaly detector trained with HSC OE. For each normal definition, a different detector and counterfactual generator was trained. In each subfigure, the first row shows anomalies, the other two corresponding counterfactuals for two different concepts. Each column is labeled with the corresponding combined normal class at the top.

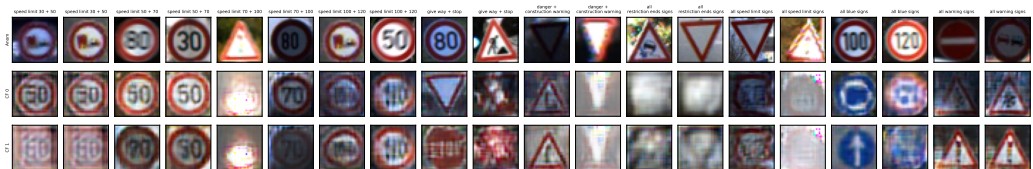

Figure 29: CEs for GTSDB and an anomaly detector trained with DSVDD. For each normal definition, a different detector and counterfactual generator was trained. In each subfigure, the first row shows anomalies, the other two corresponding counterfactuals for two different concepts. Each column is labeled with the corresponding combined normal class at the top.

