# OpenReview forum: "Anomaly Detection Exposed: Imagining Anomalies Were Normal"
_ICLR.cc/2025/Conference — ICLR 2025 Conference Withdrawn Submission_

### Official Review · Reviewer_vmsm · 2024-10-30

**Soundness:** 2
**Presentation:** 3
**Contribution:** 3
**Rating:** 5
**Confidence:** 3

**Summary:**

This paper applies counterfactual explanation (CE) to semantic image-based anomaly detection, proposing a well-designed method to generate conditional normal data, thereby providing interpretative analysis of anomaly detection results.

**Strengths:**

The article addresses a valuable yet underexplored problem.

The experimental design in the paper is extensive, and the theoretical analysis is also relatively thorough.

**Weaknesses:**

Since the paper claims to be the first to apply CE to anomaly detection, it should dedicate more space in the introduction to provide an overview of CE, or move the description of CE in the contributions section to an earlier position to make the writing in the contributions section more concise.

The paper lacks an overall framework diagram to help readers intuitively understand the work, especially for sections 3.1 and 3.2.

In section 4.3.2, the authors state that the proposed data synthesis method can effectively aid in training an AD model. However, to verify this, an appropriate experimental setup would involve a comparison with the original model that does not use the data synthesis method, rather than just comparing against a threshold of 0.5.

**Questions:**

see in Weaknesses.

---

### Official Review · Reviewer_mztZ · 2024-10-30

**Soundness:** 3
**Presentation:** 2
**Contribution:** 2
**Rating:** 3
**Confidence:** 3

**Summary:**

The paper proposes an explanation of the anomalies based on concepts. The explanation is based on a counterfactual, which is aligned with concept. In other words the counterfactual is not arbitrary, but it is aligned with some property understandable by human (authors use an example color).

**Strengths:**

* Explanation of anomaly detection is an important topic, because it can decrease the time needed to investigate the anomaly.
* Aligning explanations with concepts to which humans understand improves the acceptance of the explanation by humans.
* The method is tested on different problems, but all of them are image-based data.

**Weaknesses:**

* I miss in the motivation an explanation of a difference between anomaly detector and classifier. Because while they are trained differently, on the end anomaly detection and classifier are both functions classifying into two (or more) classes. This should be used to motivate development of new method.
* Anomaly detection systems are usually used in scenarios, where the type of anomalies is not known.  For example many intrusion and fraud detection systems have some anomaly detector to detect anomalous samples, which are rare and might be attacks. But the proposed system assumes that these anomalies would be aligned with concepts. How realistic is this assumption? The knowledge of concepts implies that the user know a lot about the domain.
* The examples of explanation raises doubts the system is useful. When an explanation of the traffic sign "yield" is a trafic sign limiting the speed to 30km/h, I do not know, what kind of information the system has given me. To me, this is not different to other traffic sign limiting speed. The same criticism go to other examples. But, the example of zebra (Figure 6) is convincing.

**Questions:**

* How the anomaly detection system differs from the classifier from the point of view of explanation?
* What is the difference in the optimization problem solved during search for the counterfactual between classifier and anomaly detection?
* What would happen, when the anomaly is difficult to explain using concepts used during training of the system? Is it possible to detect such occurrence?

---

### Official Review · Reviewer_VjpT · 2024-11-04

**Soundness:** 1
**Presentation:** 2
**Contribution:** 2
**Rating:** 3
**Confidence:** 4

**Summary:**

The paper studies an interesting problem - how to supply a counterfactual explanation to image anomaly detection problems. The authors aim to optimize an image, similar to the given abnormal image, but normal. The method aims to produce a plurality of images, exploring disentangled ways in which the image can become normal. The author suggests a loss term for that end, based on a given classifier, and explores a variety of datasets and methods.

**Strengths:**

1. The paper studies an important problem and motivates it well.
2. A variety of datasets and methods are examined
3. The results are analyzed in many relevant aspects (abnormality, disengagement, image quality, and qualitative samples)

**Weaknesses:**

1. Some parts of the manuscript are hard to follow. For example, the part in lines 130 to 189 contains a lot of notation which is not defined in the same place. Following the equation in line 181 (with x, x_bar, and x_tilde) was especially confusing, and the full motivation for using this form specifically is still not clear to me.
2. While the paper claims to report results using state-of-the-art methods, the reported methods are 2-6 years old. Especially, improved methods over DSVDD significantly are overlooked [1]. Yes, DSVDD is declared state-of-the-art.
3. Somewhat related to (2) above, standard high-resolution datasets for anomaly detection are not reported (see [1] for example)
4. While the paper claims to generate disentangled samples, in many cases the connection to the original image is not clear, and the results seem similar to a random normal image (or a disentangled set of random normal images. See Figure 4).
5. The claim in 4.3.2 ("THE COUNTERFACTUALS CAN BE USED TO TRAIN AN ANOMALY DETECTOR") is not clear. Why would that be beneficial over using the given normal train set?
6. The claim in 4.4 is not clear as well. How is the biased reported expand the one already reported in [2]?
7.  The reported disentanglement is part of the optimized loss. Can an independent measure of disentanglement be provided?
8. The disentangled variation is not necessarily meaningful. See for example Fig. 6.

Minor comments:
9. Line 400 - "Figure shows 7:


[1] Cohen, Matan Jacob, and Shai Avidan. "Transformaly-two (feature spaces) are better than one." Proceedings of the IEEE/CVF Conference on Computer Vision and Pattern Recognition. 2022.
[2] Cohen, Niv, Jonathan Kahana, and Yedid Hoshen. "Red PANDA: Disambiguating image anomaly detection by removing nuisance factors." The Eleventh International Conference on Learning Representations. 2023?

**Questions:**

Please see Weakness (2) above - Could your method work with state-of-the-art detector?
Please see Weakness (4) above
Please see Weakness (5) above
Please see Weakness (6) above
Please see Weakness (7) above

10. Are all the loss terms ablated? If so, please refer to the relevant appendix from the main text (and in the rebuttal)
11. How would your method preform on a a normal image?
12. Would the method work on the MvTec dataset for example?

---

### Official Review · Reviewer_igwr · 2024-11-04

**Soundness:** 2
**Presentation:** 3
**Contribution:** 1
**Rating:** 3
**Confidence:** 4

**Summary:**

In this paper, the authors present a new explanation method for deep learning-based image anomaly detectors. Unlike previous methods that highlight anomalous image regions, the proposed method generates counterfactual examples by modifying anomalous samples so that they are recognized as normal by the detector. This allows explanation on a semantic level by presenting "what-if" scenarios that reveal the aspects that trigger the anomaly detector. The method improves the interpretability of the anomaly detector, which is critical in safety-sensitive domains.

**Strengths:**

- The approach of providing counterfactual examples instead of highlighting anomalous regions in image anomaly detection is interesting.
- The paper clearly presents each part of the method and describes the details of the experiments.
- The authors conducted various experiments and presented the results.

**Weaknesses:**

- One of the two main weaknesses is that the proposed method is not novel enough. The proposed method simply combines anomaly detection methods with existing research on outlier exposure (OE), DISSECT, and DiffEdit with very little modification. It is difficult to argue that the authors have made a sufficient contribution to the use of existing DISSECT in the anomaly detection domain.
- The other main weakness is that the results of the presented experiments are not sufficient to prove the performance of the method, and I cannot agree with the authors' analysis of the results.
    - Among the qualitative results, Colored-MNIST and GTSDB show understandable results, but for datasets such as MNIST and CIFAR-10, I am not convinced that the method provides an “explanation”. Without the main text, it's hard to recognize what is normal when looking at the result images.
    - Even in the quantitative results, many results do not show that the method performs well enough. For example, when training the anomaly detector on the counterfactual examples generated in Table 2, the authors claim that it performs well because the AUROC is higher than the random chance of 50%, which is generally unacceptable. Also, the results in Table 1 and Table 3 are not consistent (e.g., the results for CIFAR-10 are good enough in Table 1, but are completely destroyed in Table 3).
    - The proposed method does not seem to work for detectors like DSVDD. The authors state in section 3.3 that their proposed method can be applied to “any anomaly detector that produces real-valued anomaly scores”, but based on their results, it seems to work only on anomaly detectors that perform “well enough”.
- (Minor) The paper is not polished enough.
    - There are too many citation formatting errors to be treated as typos (L153, L214, L234-236, ...). The authors should review the entire paper and correct all of these formatting errors.
    - There are some misstatements in the citations. For example, PaDiM (Defard et al., 2021) and PatchCore (Roth et al., 2022) in L89 use pre-trained networks and are not transfer learning methods.

**Questions:**

- What is the computational complexity of the proposed method? It seems that the added GAN and diffusion model would add significant computational cost.
- How should the proposed method be used in situations where there is no outlier dataset for OE that is sufficiently similar to the target dataset, and what do you think the performance will be in such situations?
- It would be helpful to have a diagram of the proposed method to better understand the overall structure.
- (Minor) Area Under the ROC curve as AuROC is unnatural. AUROC would be better.
- (Minor) Related works should be re-categorized or paragraph titles should be revised.

---

### Note · Authors · 2024-11-21

I have read and agree with the venue's withdrawal policy on behalf of myself and my co-authors.